# Optical imaging of metabolic dynamics in animals

Lingyan Shi [1], Chaogu Zheng[2], Yihui Shen[1], Zhixing Chen[1], Edilson S. Silveira[1], Luyuan Zhang[1], Mian Wei[1], Chang Liu[1], Carmen de Sena-Tomas[3], Kimara Targoff[3] & Wei Min [1,4]

Direct visualization of metabolic dynamics in living animals with high spatial and temporal resolution is essential to understanding many biological processes. Here we introduce a platform that combines deuterium oxide ($D_2O$) probing with stimulated Raman scattering (DO-SRS) microscopy to image in situ metabolic activities. Enzymatic incorporation of $D_2O$-derived deuterium into macromolecules generates carbon–deuterium (C–D) bonds, which track biosynthesis in tissues and can be imaged by SRS in situ. Within the broad vibrational spectra of C–D bonds, we discover lipid-, protein-, and DNA-specific Raman shifts and develop spectral unmixing methods to obtain C–D signals with macromolecular selectivity. DO-SRS microscopy enables us to probe de novo lipogenesis in animals, image protein biosynthesis without tissue bias, and simultaneously visualize lipid and protein metabolism and reveal their different dynamics. DO-SRS microscopy, being noninvasive, universally applicable, and cost-effective, can be adapted to a broad range of biological systems to study development, tissue homeostasis, aging, and tumor heterogeneity.

[1] Department of Chemistry, Columbia University, New York, NY 10027, USA. [2] Department of Biological Sciences, Columbia University, New York, NY 10027, USA. [3] Department of Pediatrics, Columbia University, New York, NY 10027, USA. [4] Kavli Institute for Brain Science, Columbia University, New York, NY 10027, USA. These authors contributed equally: Lingyan Shi, Chaogu Zheng.  Correspondence and requests for materials should be addressed to W.M. (email: wm2256@columbia.edu)

Understanding the dynamics of metabolism in multicellular organisms is important to unraveling the mechanistic basis of many biological processes. Although metabolomic technologies can catalog thousands of metabolites residing in cells, nondestructive tools are limited for in situ visualization of metabolic activities, such as protein and lipid synthesis and degradation, at subcellular resolution in living organisms. Magnetic resonance spectroscopic imaging and positron emission tomography can provide metabolic information noninvasively and have wide oncological application but lack sufficient spatial resolution[1]. Microautoradiography and fluorescence microscopy can visualize metabolism at the single-cell level but require radioactive and fluorescent labeling of the substrate, respectively; these labeling are often toxic to cells and often perturb the native metabolic processes[2]. Nanoscale secondary ion mass spectrometry and the more recent multi-isotope imaging mass spectrometry can measure the incorporation of nontoxic stable isotopes, like $^{15}N$ and $^{13}C$, at submicrometer resolution and spatially track the labeling of biomolecules; but both methods are destructive to living tissues and have limited resolvability for macromolecules[3,4].

Here we developed a general method that combines deuterium oxide probing and stimulated Raman scattering (DO-SRS) microscopy to provide imaging contrast for visualizing metabolic dynamics in situ. Through systematic investigation of the carbon–deuterium (C–D) vibrational spectrum, we discovered Raman shifts associated with C–D bond-containing lipids, proteins, and DNA, respectively, and further revealed that this spectral selectivity resulted from the sparse labeling pattern and inherently different chemical environments surrounding the C–D bond in different types of macromolecules. By applying DO-SRS microscopy to living cells and animals, we not only demonstrated its broad utility, high sensitivity, noninvasiveness, subcellular resolution, compatibility with other imaging modality, and suitability for in vivo live imaging in mammals but also gained new insights on the metabolic basis of several biological processes.

## Results

**$D_2O$ is an ideal contrast agent for metabolic activities.** Water ($H_2O$), the ubiquitous solvent of life, diffuses freely across cell and organelle membranes and participates in the vast majority of biochemical reactions. As an isotopologue of water, heavy water ($D_2O$) can rapidly equilibrate with total body water in all cells within an organism and label cellular biomolecules with deuterium (D) by forming a variety of X–D bonds through non-enzymatic H/D exchange and enzymatic incorporation (Fig. 1a). The former is spontaneous and reversibly forms oxygen–deuterium (O–D), nitrogen–deuterium (N–D), and sulfur–deuterium (S–D) bonds on existing molecules, whereas the latter depends on enzyme-catalyzed chemical transformation that irreversibly breaks the O–D bond and forms C–D bonds on newly synthesized molecules[5]. Through such transformation, deuterium quickly labels the metabolic precursors, such as non-essential amino acids (NEAAs), acetyl-CoA, and deoxyribose, which are then slowly incorporated into proteins, lipids, and DNA, respectively[6–8] (Fig. 1a). As often with the rate-limiting step, the synthesis of C–D bond-containing macromolecules from the precursors is governed by cellular metabolic activities. Therefore, $D_2O$ can be used as a universal probe to track metabolic rate through the emergence of C–D bond-containing macromolecules (hereinafter referred to as D-labeled macromolecules).

Raman spectroscopy provides a noninvasive, optical approach to distinguish metabolic incorporation from non-enzymatic exchange in situ, because various X–D bonds have intrinsically distinct stretching vibrational features. We found that the Raman spectrum of C–D bond was clearly separated from those of C–H, O–D in $D_2O$, and the non-enzymatically formed O–D, S–D, and N–D bonds (Fig. 1b), which allows the direct detection of biosynthetic incorporation of deuterium through the amount of C–D bonds. In fact, spontaneous Raman microspectroscopy has recently been employed to identify metabolic activity in bacteria after $D_2O$ treatment[9,10], but this approach has difficulties in generating spatially resolved images due to low sensitivity and slow imaging speed. Compared to spontaneous Raman spectroscopy, SRS microscopy is an emerging nonlinear Raman imaging technology with substantial sensitivity boost through quantum amplification by stimulated emission, which enables at least three orders of magnitude faster acquisition time, fine spectral resolution, compatibility with fluorescence, and three-dimensional (3D) optical sectioning capability in tissues and even living animals[11–14]. These unique advantages of SRS microscopy, combined with our new discoveries of the chemical features of the C–D vibrational spectrum (described below), led to the development of DO-SRS microscopy, which uses $D_2O$ as a imaging contrast agent to specifically trace lipid, protein, and DNA metabolism in cells and tissues.

We first demonstrated DO-SRS microscopic imaging of metabolism in cultured cells. By treating HeLa cells with medium containing 70% $D_2O$ for 24 h (toxicity only arises when $D_2O$ concentration exceeded 80%; Supplementary Fig. 1) and then tuning SRS microscopy to target the C–D frequency, we found that C–D signal was undetectable at the beginning of the treatment but increased dramatically in all cells at 24 h (Fig. 1b). This result confirms that C–D signal specifically and effectively reports newly synthesized molecules, whereas C–H signal represents pre-existing pool of molecules. Importantly, the separation between the O–D peak and the C–D peak means that C–D signal is essentially free of interference from the overwhelming O–D background, thus washing off the $D_2O$ probe before imaging is unnecessary.

**The sensitivity and in vivo live imaging capacity of DO-SRS microscopy.** In humans, deuterium is widely used as a stable isotope to measure body composition and metabolic rate[15–17]. Daily intake of 60–70 ml $D_2O$, which results in ~2% body water enrichment, does not cause any adverse symptoms[18,19], and is considered to be safe. We found that a comparable level (2.4–2.8%) of enrichment in mice by administration of 4% $D_2O$ as drinking water (the dilution of body water relative to drinking water is typically 30–40% in rodents[19]) produced easily detectable C–D signals in sebaceous glands (Fig. 1c).

In mice, $D_2O$ enrichment below 20% did not cause any effect on physiological processes, including no acute adverse events, no effects on cell division in all major cell renewal systems, no perturbation on physiology, growth, appetite, and reproduction, and no teratogenic effects, even in multigenerational studies[17,20,21]. Thus, we gave mice 25% $D_2O$ as drinking water to achieve a safe level (15–17.5%) of enrichment in body water and obtained a 6-fold increase in C–D signal compared to that from mice drinking 4% $D_2O$ (Fig. 1c). The nearly linear relationship between $D_2O$ enrichment and signal intensity allowed us to extrapolate the detection limit (Fig. 1c). Administration of 1.4% $D_2O$ (~0.9% enrichment) is sufficient to achieve a signal-to-noise ratio (S/N) of 10, and for S/N higher than 2, administration of 0.3% $D_2O$ would be needed. Although detectable signal can be generated at low $D_2O$ enrichment, we chose ~16% $D_2O$ enrichment (from oral administration of 25% $D_2O$) in the following mouse studies in order to achieve large dynamic range and also reveal low biosynthetic activity within a short time frame.

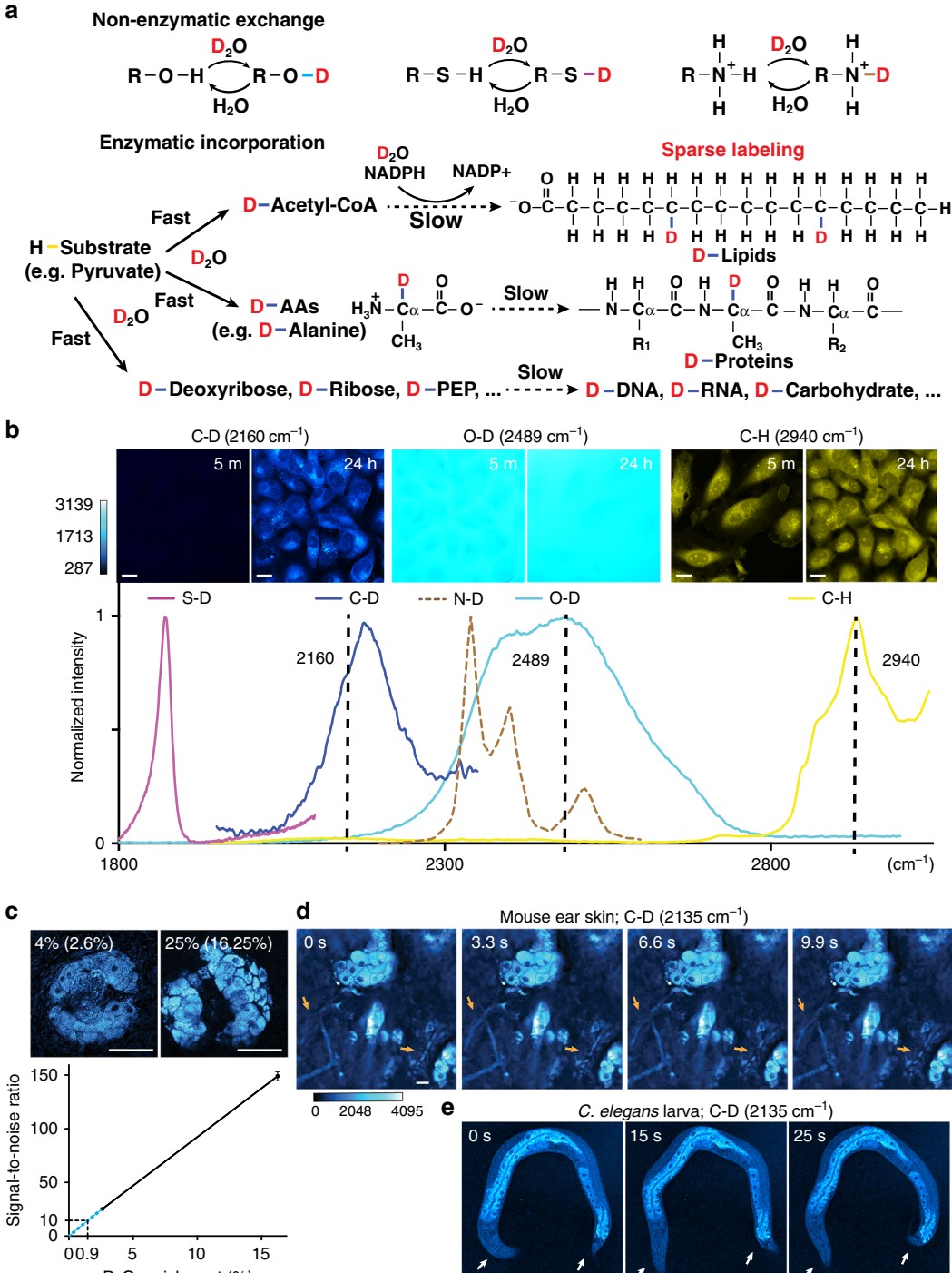

**Fig. 1** SRS microscopic imaging of biosynthetic incorporation of deuterium from $D_2O$ into macromolecules in living cells and animals. **a** $D_2O$-derived deuterium can form O–D, S–D, and N–D bonds through reversible non-enzymatic H/D exchange and be incorporated into C–D bonds of metabolic precursors for the synthesis of macromolecules through irreversible enzymatic incorporation. Note that, under the condition of sparse labeling, the position of deuterium labeling on lipids is random, and this illustration only represents one possible labeling pattern. **b** Various X–D bonds produced Raman peaks at distinct positions. C–D, C–H, and O–D spectra were collected from Hela cells grown in 70% $D_2O$ medium, S–D spectrum from saturated cysteine dissolved in $D_2O$, and N–D spectrum adopted[65]. SRS microscopic images were collected for C–D, O–D, and C–H signal of HeLa cells at 5 min and 24 h after adding $D_2O$-containing medium. **c** Signal-to-noise ratio (S/N; noise = 1 μV) for SRS microscopic signal of sebaceous glands at 2135 cm$^{-1}$ from mice that drank 4% or 25% $D_2O$ for 8 days. Percentages of $D_2O$ enrichment in body water are shown in parentheses. Detection limit at S/N = 10 or 2 was calculated based on the linear relationship between average S/N and $D_2O$ enrichment level. **d**, **e** Frames from live SRS microscopic imaging recordings of the sebaceous glands under the ear skin of intact mice that drank 25% $D_2O$ for 9 days (Supplementary Movie 1–3) and living fourth stage *C. elegans* larva that grew on 20% $D_2O$-containing NGM plates for 4 h (Supplementary Movie 4–6). The blood flow (orange arrows) in mouse visualized with two-photon absorption contrast and the movement of the *C. elegans* body (white arrows) indicated that the animals under imaging were alive. Scale bar = 20 μm. C–D signal was shown in cyan hot LUT, and color scale bars display the scale for the lookup table values

The wash-free aspect of DO-SRS microscopy enables us to image the real-time C–D signal dynamics of live cells and intact animals in the presence of $D_2O$. We performed in vivo live imaging of the sebaceous glands under the ear skin of anesthetized mice that drank 25% $D_2O$ for 9 days. Leveraging the 3D optical sectioning and substantial imaging depth of nonlinear excitation of SRS microscopy (about hundreds of microns), we observed clear C–D signal in sebocytes in living mice (Fig. 1d and Supplementary Movie 1–3). To our knowledge, this is the first time that deuterium-labeled macromolecules have been visualized in intact living mammals, thanks to the high-speed live imaging capability of SRS microscopy. At a whole-organism level, we were able to monitor C–D signals in moving *Caenorhabditis elegans* larva growing in a 20% $D_2O$ environment (Fig. 1e and Supplementary Movie 4–5). In *C. elegans*, <60% $D_2O$ concentration had no observable toxicity and did not affect cell division (Supplementary Fig. 1). With the above proof of concept, we show that the safeness of $D_2O$ as a probe, combined with the sensitivity and penetration of SRS microscopy as a detection mechanism, would allow for long-term in vivo tracking of metabolic activities.

**Macromolecule-specific C–D vibrational frequencies.** Because $D_2O$ is a universal probe and enables the labeling of different types of macromolecules, the C–D signal generated from $D_2O$ probing is considered to report the total metabolic activity. Whether sufficiently distinct C–D vibrational frequencies exist to allow the separation of different types of D-labeled macromolecules is unaddressed.

To address this question, one might attempt to translate the spectral knowledge of C–H region to the C–D region, but we reasoned that this translation does not apply. C–H stretching vibrational spectra contain a main peak at 2940 $cm^{-1}$ (denoted as $CH_P$ channel) originating from protein-related $CH_3$ stretching, a main peak at 2845 $cm^{-1}$ ($CH_L$ channel) from lipid-related $CH_2$ stretching, and a shoulder peak at 2967 $cm^{-1}$ ($CH_{DNA}$ channel) from DNA-related C–H stretching[22–24]. However, the C–H spectral distinction (based on the number of covalent hydrogen atoms) cannot be simply translated to C–D stretching, because $CD_2$ and $CD_3$ groups would rarely exist under relatively low $D_2O$ enrichment (15–17.5%) in body water. Such low $D_2O$ concentration means that deuterium could only sparsely label the newly synthesized macromolecules (Fig. 1a), and the C–D vibrational signal would be dominated by CD mode (only one carbon-bonded H atom is replaced by D atom).

To dissect the spectral features of C–D vibrational modes upon $D_2O$ treatment, we first scanned the C–D region (2109–2210 $cm^{-1}$), tested 5 Raman shifts with equal intervals, and found identifiable difference among the 5 images (Fig. 2a). C–D images acquired at lower wavenumbers (especially 2135 $cm^{-1}$) resembled the cellular distribution patterns of lipids from the $CH_L$ channel, and images acquired at higher wavenumbers (especially 2185 $cm^{-1}$) resembled the protein signal from $CH_P$ channel in all tissues we tested, including fibroblast-like COS7 cells, *C. elegans* larva, zebrafish embryos, and mouse tissues (Fig. 2a). Thus, we hypothesized the signal at 2135 $cm^{-1}$ may represent newly synthesized D-labeled lipids and the signal at 2185 $cm^{-1}$ as D-labeled proteins.

To confirm this hypothesis, we found that inhibiting protein synthesis with anisomycin in $D_2O$-treated HeLa cells led to a peak centered at 2135 $cm^{-1}$, representing the remaining D-labeled lipid signal, and blocking lipid synthesis with fatty acid synthase inhibitor TVB-3166 led to a peak centered at 2185 $cm^{-1}$, representing the remaining D-labeled protein signal (Fig. 2b). The sum of the signals from the two treatments appeared similar to the control group treated only with $D_2O$. At the tissue level,

removing proteins by proteinase K treatment in mouse tissues abolished the peak at 2185 $cm^{-1}$, and dissolving lipids using methanol abolished the peak at 2135 $cm^{-1}$ (Fig. 2c). Applying methanol wash to multiple mouse tissues from various organs, we obtained the average spectra for D-labeled lipids (signal reduction by methanol wash) and proteins (residual signal after methanol wash), consistently showing peaks at the two frequencies (Fig. 2d). The above data confirmed macromolecule-specific Raman shifts for D-labeled lipids and proteins at 2135 $cm^{-1}$ (denoted as $CD_L$ channel) and 2185 $cm^{-1}$ ($CD_P$ channel), respectively.

We then sought for the chemical basis of spectral distinction in CD vibration by comparing D-labeled lipid and protein spectra with assigned CD vibrational modes in model compounds. The peak of D-labeled lipid at ~2140 $cm^{-1}$ matched well with the singly deuterated C–D stretching in *12-D1*-palmic acid but not the $CD_2$ symmetric stretching mode at ~2100 $cm^{-1}$ in perdeuterated palmitic acid (*D31-*) (Fig. 2e), supporting the idea of sparse labeling (Fig. 1a). Similarly, the D-labeled protein peak around 2185 $cm^{-1}$ matched the peak for $C(\alpha)$-D vibration in *D4*-alanine but not the other three peaks assigned to the side chain $CD_3$ group[25] (Fig. 2f). This pattern also supported sparse labeling and, more importantly, indicated that most of the deuterium labeling in newly synthesized proteins occurred at the $\alpha$ carbon position through reversible transamination of free AAs[26] (Fig. 1a).

The above evidence suggests that the underlying principle for the observed spectral separation of D-labeled lipids (around 2140 $cm^{-1}$) and proteins (around 2185 $cm^{-1}$) is the inherently distinct chemical environments of the constituting fatty acids and amino acids—deuterium-bonded carbon atoms in a hydrocarbon chain of lipids and the main chain of polypeptides are connected to chemical groups with different polarities. We next asked whether such principle can be extended to other macromolecules. By chemically isolating the major macromolecules (lipids, proteins, and DNA) from $D_2O$-treated HeLa cells, we found that the Raman spectra of D-labeled lipids and proteins matched well with the corresponding spectra obtained in situ, as expected, and that the Raman peak of D-labeled DNA was blue-shifted compared to D-labeled proteins (Fig. 2g). Deuterium labels DNA with higher chance at the C1′ and C2′ positions on the deoxyribose[19,27], and the blue shift of DNA's CD peak may be attributed to the fact that more carbon atoms in deoxyribose are bonded to electronegative oxygen atoms. Based on the spectra of the cellular extracts, we chose to image DNA at 2210 $cm^{-1}$ (designated as $CD_{DNA}$ channel).

**Spectral unmixing of signals for D-labeled macromolecules.** Although the peaks of CD signals in lipids, proteins, and DNA are separated, their overall spectra overlap substantially (Fig. 2d, g). Thus we next developed a three-component unmixing algorithm to computationally decompose the mixed CD signals into three macromolecule-specific elements (see Supplementary Note 1 and Methods for details). Applying the method to dividing cells, we successfully separated D-labeled lipids, proteins, and DNA, removing the bleed-through signal for each channel (Fig. 3a). Since $CD_{DNA}$ signal was very weak in non-dividing cells, we only needed to unmix $CD_L$ and $CD_P$ signals using a simplified two-component equation in most experiments. The unmixing effectively removed residual bleed-through signals in $CD_L$ and $CD_P$ channels, revealing the genuine distribution of D-labeled lipids and proteins (Fig. 3, Supplementary Fig. 2, and Supplementary Note 1).

Overall, our unmixing technique enables, for the first time, in situ deconvolution of D-labeled lipids and proteins signal via SRS microscopy. It is worth noting the difference between the unmixing method we developed and the label-free counterpart

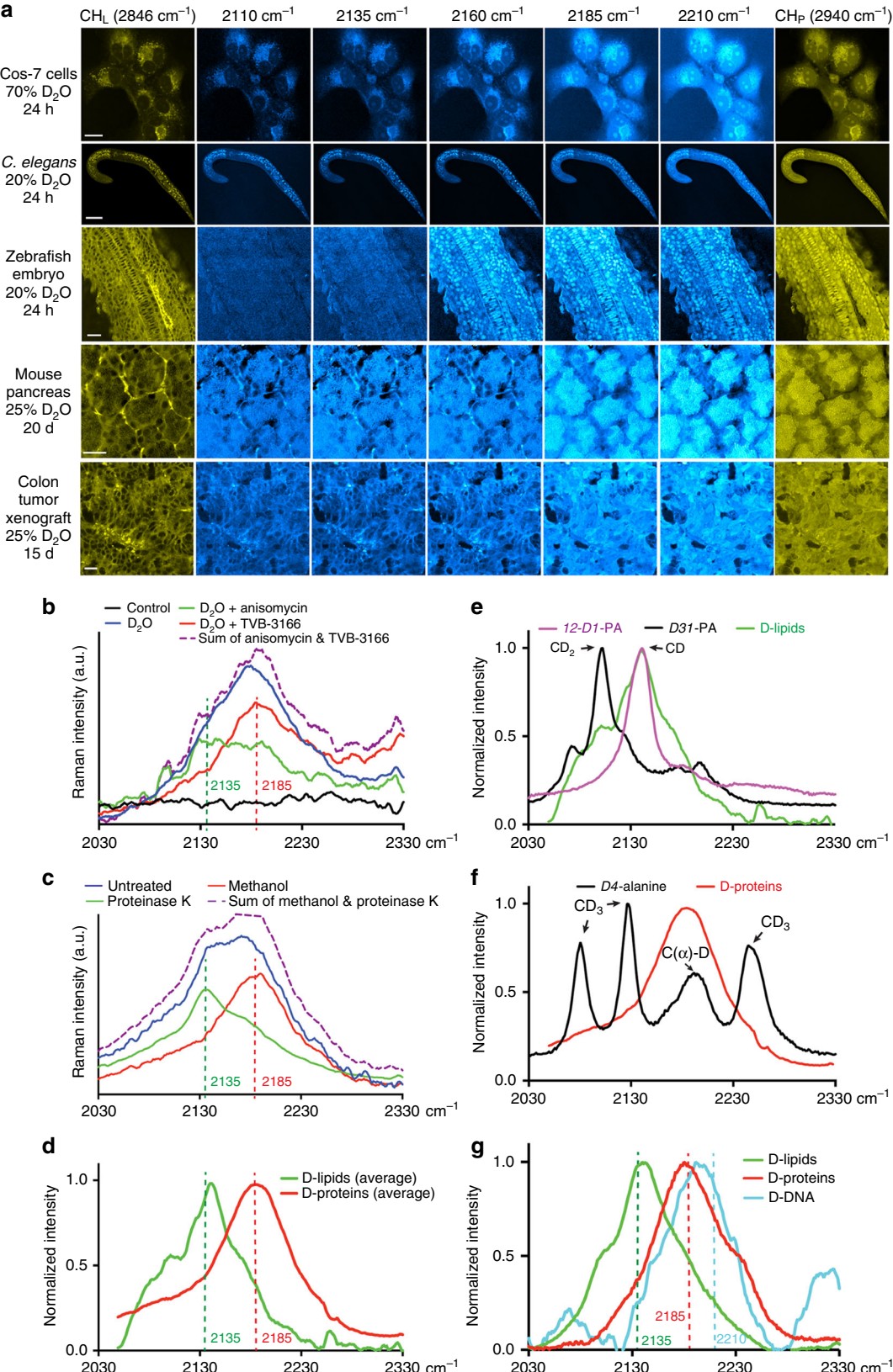

previously reported. Unlike the unmixing of $CH_L$ and $CH_P$, which used pure standard compounds (oleic acid for lipids and bovine serum albumin (BSA) for proteins)[22,23], our unmixing method is tailored for sparse labeling pattern of deuterium incorporation in vivo. Without commercially available standards for randomly, sparsely deuterated lipids and proteins, we generated standards using either chemically isolated or in situ D-labeled lipids and proteins to determine the unmixing coefficients. The ability to separate the signals for D-labeled lipids and proteins enabled simultaneous visualization of the metabolic dynamics of lipid and

**Fig. 2** Identification of specific Raman shifts with macromolecular selectivity within the broad C–D vibrational spectra. **a** SRS microscopic images of various cells and tissues from animals treated with $D_2O$ for the indicated amounts of time. Images were collected using previously known Raman shifts for CH-containing lipids ($CH_L$ 2846 cm$^{-1}$) and proteins ($CH_P$ 2940 cm$^{-1}$) and five wavenumbers (2110, 2135, 2160, 2185, and 2210 cm$^{-1}$) within the C–D broadband. Scale bar = 20 µm. **b** High wavenumber Raman spectra (2030–2330 cm$^{-1}$) of Hela cells grown in DMEM made of 70% $D_2O$ in the absence or presence of fatty acid synthase inhibitor TVB-3166 or protein synthesis inhibitor anisomycin. Cells grown in DMEM made of 100% $H_2O$ were used as control (black). Purple curve shows the sum of the values on the green and red curves. **c** High wavenumber Raman spectra of deuterium-labeled xenograft colon tumor tissues treated with protease K or washed with methanol for 24 h. Mice bearing the xenograft drank 25% $D_2O$ as drinking water for 15 days before tumor tissues were harvested and imaged. **d** The normalized Raman spectra of tissue after 24 h methanol wash (D-labeled protein signal in red) and the difference spectra before and after methanol wash (D-labeled lipid signal in green), averaged from various mouse tissues. **e** Comparison of Raman spectra of 12-D1-palmitic acid (100 mM dissolved in DMSO), D31-palmitic acid (100 mM in DMSO), and in situ D-labeled lipid standards. **f** Comparison of Raman spectra of D4-alanine (100 mM in PBS) and D-labeled protein standards. Assignments of the peaks were made according to a previous report[25]. **g** Raman spectra of biochemically extracted lipids, proteins, and DNA from HeLa cells grown in DMEM media containing 70% $D_2O$

protein in the same tissue, which is important for addressing many fundamental questions about the different pathways of cellular metabolism.

**Optical imaging of de novo lipogenesis via DO-SRS microscopy.** Previously, our laboratory and others have developed methods to visualize lipogenic activities in living tissues by supplying deuterium-labeled fatty acids (D-FAs), such as palmitic acid, oleic acid, and arachidonic acid, and imaging C–D bonds in newly synthesized lipids[28–31]. However, DO-SRS microscopy is fundamentally different from those previous methods because $D_2O$ is a noninterfering probe that does not affect native metabolism and is a non-carbon tracer that can probe activities of de novo lipogenesis.

D-FAs are known to be taken up by cells through scavenger pathways and then incorporated directly into lipids, whereas $D_2O$ freely diffuses into cells and labels newly synthesized lipids through de novo lipogenesis. Moreover, the dependence on cellular uptake for D-FAs can also result in bias among various cell types, which does not occur for $D_2O$ probing. The different effects of these two types of probes are clearly evident in cultured cells. For example, supplementation of D-labeled palmitic acids (D-PA) or oleic acids (D-OA) in HeLa cells led to the accumulation of $CD_L$ signal in lipid droplets (Fig. 4a), whereas $D_2O$ probing generated very few lipid droplets in both $CD_L$ and $CH_L$ channels and produced much more uniform $CD_L$ signal in cytoplasmic membrane structure.

The large number of lipid droplets caused by the treatment of D-FAs even at low concentrations (10 µM) suggests that exogenous fatty acids likely perturbed native cellular metabolism. Moreover, different types of fatty acids altered lipid metabolism in distinct ways; for the same HeLa cells, treatment of 10 µM D-OA generated remarkably more lipid droplets than D-PA (Fig. 4a). Our earlier study also found that PA but not the unsaturated OA, when applied at higher concentration (~200 µM), drove the formation of solid-like microdomain and membrane phase separation in endoplasmic reticulum (ER)[30]. Thus, compared to D-FAs, $D_2O$, as a metabolic bystander, would be a better probe for monitoring endogenous lipogenesis in general.

**DO-SRS microscopy tracks lipid metabolism in C. elegans.** In animals, we first applied DO-SRS microscopic imaging to assess how much de novo synthesis contributes to the production of total lipids in C. elegans, which relies on both dietary uptake (from food source E. coli bacteria) and de novo lipogenesis for total lipid synthesis. For the quantification purpose, we developed $CD_L/CH_L$ as a ratiometric indicator for the amount of newly synthesized lipids normalized against variations among individuals and heterogeneity within the same tissue (see Methods). We

found that, when both grown on 20% $D_2O$ plates, animals fed on non-labeled bacteria (grown in $H_2O$) had much lower $CD_L/CH_L$ ratio (~3.1%) than animals fed on D-labeled bacteria (~22.3%; Fig. 4b and Supplementary Fig. 3A), indicating that ~14% of total lipids were synthesized de novo and the rest ~86% were incorporated or modified from E. coli fatty acids. This result agrees with previous mass spectrometric studies using dietary $^{13}C$ labeling, which found that C. elegans only de novo synthesized ~7% of palmitate and 12–19% of 18-carbon fatty acids from acetyl-CoA[32]. However, compared to mass spectrometry, which can only measure metabolic incorporation of isotopes in a bulk of thousands of worms, DO-SRS microscopic in situ imaging detects metabolic activity at the individual animal level, requires much less materials, and can reveal variations among individuals.

$D_2O$ is a better probe than D-FAs for detecting lipogenesis in C. elegans because of the following reasons. First, we directly compared $CD_L$ signals generated by $D_2O$ probing and D-FA supplementation (Supplementary Figure 3B and C) and found that, although both labeled lipid droplets in similar patterns, 20% $D_2O$ treatment produced over two-fold stronger $CD_L$ signal than 4 mM deuterated palmitic acid, the highest concentration used in previous studies[29,31]. Second, $D_2O$ is able to track the ~14% lipids generated by de novo lipogenesis, whereas D-FAs cannot. Third, because bacteria grown on $D_2O$ plates produce a variety of D-labeled fatty acids, lipids, and their metabolic intermediate, which all become dietary nutrient for C. elegans, $D_2O$ probing (by generating D-labeled bacteria) can monitor lipid synthesis more accurately and more robustly than the supplementation of a single type of D-FA. Fourth, $D_2O$ probing also generates D-labeled protein signals in addition to lipid signals, whereas D-FA treatment does not (Supplementary Fig. 3B).

DO-SRS microscopy allowed us to study the dynamics of lipid metabolism in C. elegans. When transferred from $H_2O$ plates to 20% $D_2O$ plates, normal fourth-stage larva (L4) showed newly synthesized lipid droplets in the intestine as early as 20 min; $CD_L$ signal continued to increase for 6 h and expanded into hypodermis, suggesting fast and robust lipogenesis (Fig. 4c). In contrast, when developmentally arrested dauer larva were placed onto $D_2O$ plates with food (E. coli bacteria), new lipid signals did not appear until 6 h after the transfer and took 16 h to reach the 6-h intensity of normal larva (Fig. 4d). The different dynamics of normal and dauer larva reflects the additional time required for dauers to exit the non-feeding, diapause stage and to resume metabolic activities and life cycle[33,34]. Interestingly, our data connect metabolic dynamics to previously reported changes in transcriptome during this dauer recovery process, because the onset of lipogenic activities closely follows the expression of genes involved in glycolysis, tricarboxylic acid cycle, fatty acid oxidation, and oxidative phosphorylation[34]. This result suggested that the buildup of storage fat in recovering dauers began immediately after the production of ATP from food digestion and

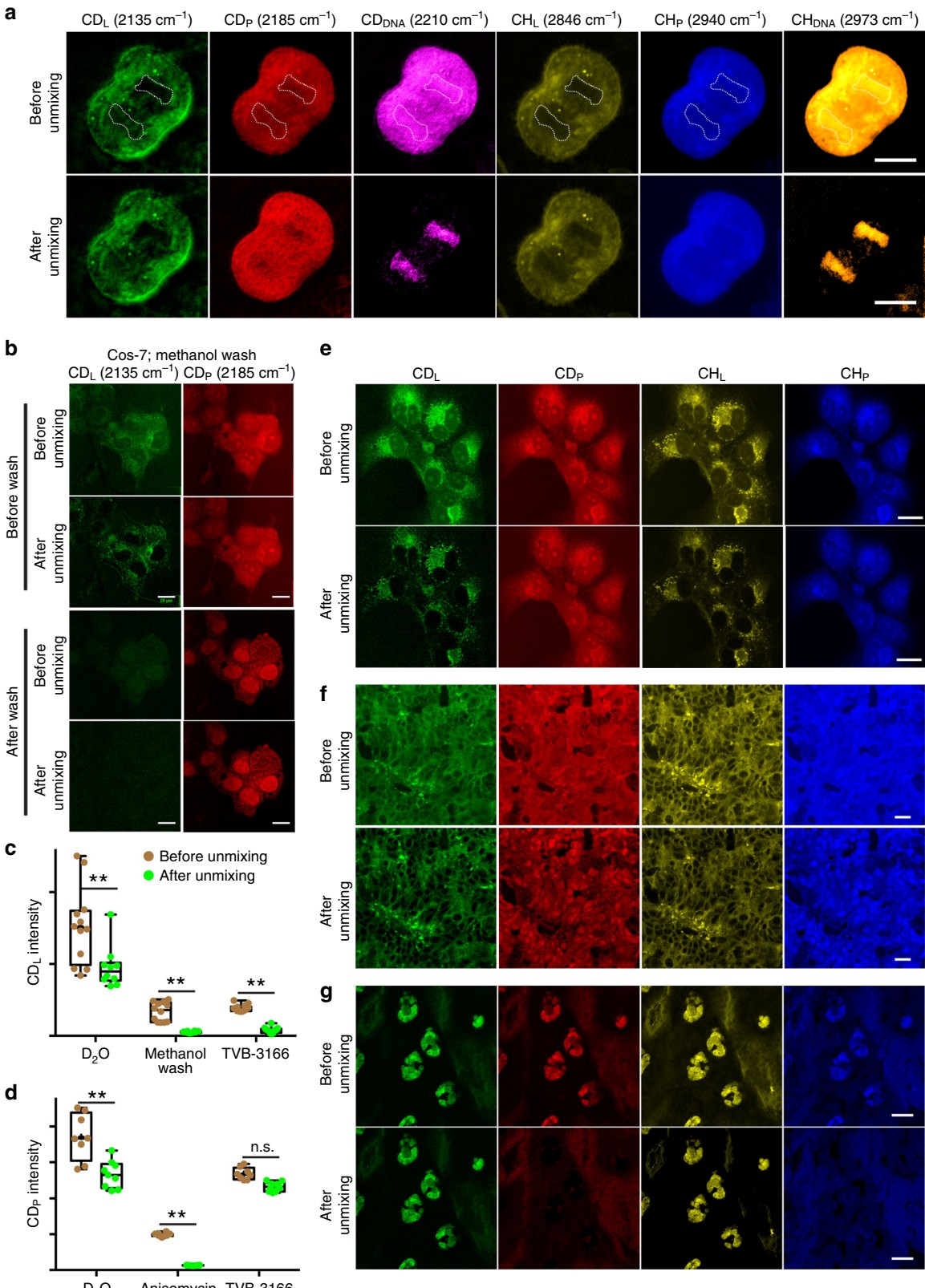

may be important for the preparation of dauer-to-L4 molt, during which food intake stopped and $CD_L$ signal stayed flat (8 to 12 h; Fig. 4d).

We can also monitor lipid degradation through a pulse-chase experiment. For example, L4 animals were first pulsed for 1 h on 20% $D_2O$ plates and then transferred back to regular $H_2O$ plates. $CD_L$ signal dropped by ~60% in the first 5 h (Fig. 4e), suggesting a quite fast lipid turnover in the developing larva. Thus DO-SRS microscopy enabled the visualization of both lipid anabolism and catabolism.

**Fig. 3** Spectral unmixing of D-labeled lipids, proteins, and DNA. **a** Separation of CD protein and DNA signals via unmixing in dividing cells (see Methods for details). Dashed outlines enclose the nuclei. In all the figures, $CD_L$, $CD_P$, $CD_{DNA}$, $CH_L$, $CH_P$, and $CH_{DNA}$ channels show signals collected at 2135, 2185, 2210, 2846, 2940, and 2973 cm$^{-1}$, respectively, and are color-coded in green, red, pink, yellow, blue, and gold, respectively. **b** SRS microscopic images, collected from the $CD_L$ and $CD_P$ channels, of COS-7 cells grown in 70% $D_2O$ DMEM for 24 h and images of the same cells after methanol wash, with or without the application of unmixing algorithm. **c**, **d** Quantification of the mean SRS microscopic intensity (mean ± s.d.) from $CD_L$ and $CD_P$ channels of COS-7 cells under various conditions before and after unmixing ($N = 12$ for each condition). Double asterisk indicates $p < 0.01$ in an unpaired $t$ test. **e**–**g** Example sets of images before and after the application of $CD_L/CD_P$ unmixing for COS-7 cells grown in 70% $D_2O$-containing DMEM for 24 h (**e**), xenograft colon tumor tissues from mice drinking 25% $D_2O$ for 15 days (**f**), and sebaceous gland tissues from mice drinking 25% $D_2O$ for 3 days (**g**). $CH_L/CH_P$ unmixing was performed according to previous studies[22]. Scale bar = 20 μm. Tissue-specific unmixing parameters can be found in Methods

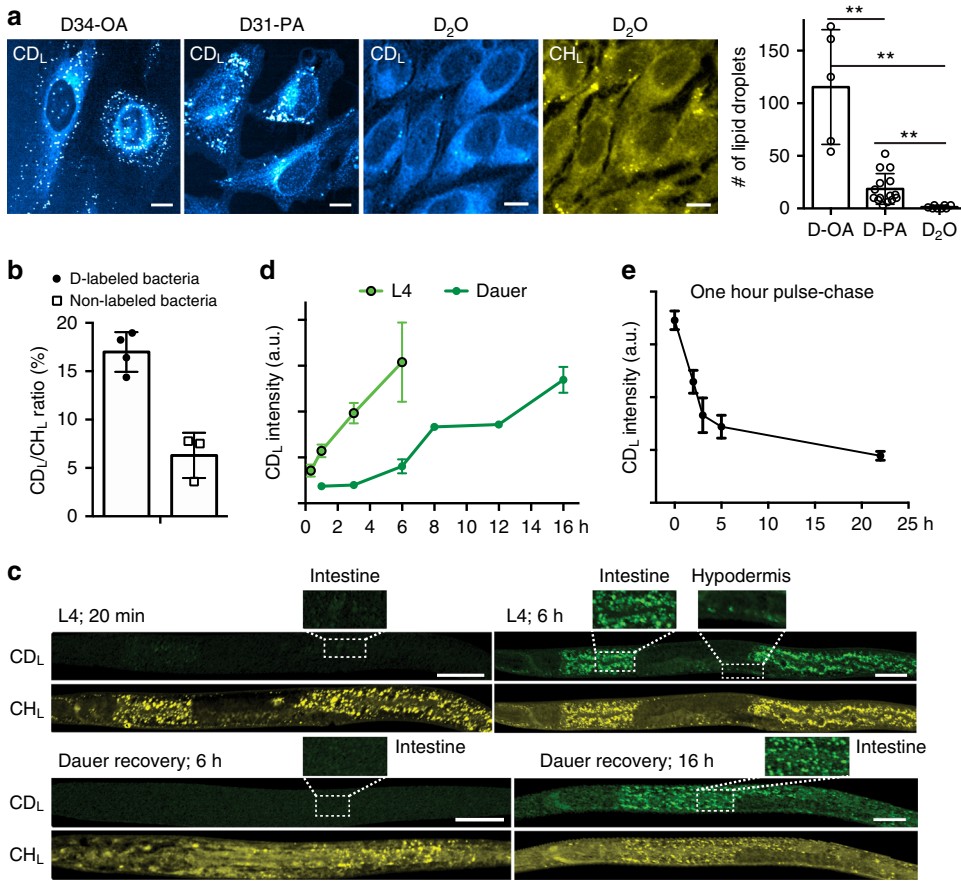

**Fig. 4** DO-SRS microscopy visualizes de novo lipogenesis in cultured cells and in *C. elegans*. **a** SRS microscopic signals at 2110 cm$^{-1}$ of HeLa cells grown in DMEM with 1% FBS and 10 μM of *D34*-oleic acids (D-OA) or *D31*-palmitic acids (D-PA) for 6 h, compared to the signals from HeLa cells grown in DMEM with 1% FBS and 70% $D_2O$ for 24 h. The numbers of lipid droplets formed in the three conditions were quantified (mean ± s.d.; $N = 5$ for D-OA, 15 for D-PA, and 8 for $D_2O$; double asterisks indicate $p < 0.01$ in an unpaired $t$ test). $CH_L$ signal for $D_2O$ treatment confirmed the presence of few lipid droplets. **b** SRS microscopic signal intensity ratios (mean ± s.d.) of third-stage larva grown from eggs on 20% $D_2O$ plates seeded with live ($N = 4$) or UV-killed ($N = 3$) *E. coli* OP50 24 h before imaging. Live bacteria were labeled by deuterium, and killed bacteria were not labeled. **c** Normal fourth-stage larva (L4) or dauer larva were transferred from regular NGM plates made of $H_2O$ to plates made of 20% $D_2O$ and then imaged at different time points. *E. coli* OP50 were seeded onto the $D_2O$ plates 24 h before the experiments. Scale bar = 20 μm. **d** Quantification of the mean intensity over the entire worm body. **e** L4 animals were first grown on $D_2O$ plates for 1 h and then transferred to $H_2O$ plates and imaged at different time points after the transfer. $CD_L$ mean intensity was plotted. For all *C. elegans* experiments, at least eight animals were imaged for each condition to calculate mean intensity; mean ± s.d. of the mean intensity is shown

## Visualizing de novo lipogenesis in mice

By adding $D_2O$ to the drinking water of mice, previous studies showed the incorporation of deuterium into the total biomass in various mouse organs but could not image and differentiate D-labeled lipids and proteins in situ[5–8]. Using DO-SRS microscopic imaging with macromolecular selectivity, we found that different mouse organs show metabolic preference to either protein or lipid biosynthesis, which reflects their different functions (Supplementary Fig. 4A). For example, lipid-rich tissues, such as sebaceous glands, myelin sheath in the brain, and adipose tissues, showed strong $CD_L$ signal and weak $CD_P$ signal (Fig. 5 and Supplementary Fig. 4A). Because the visualization of in situ lipid synthesis dynamics in mouse models had been particularly challenging with previous methods, which mostly capture the static level[35], we aimed to demonstrate the utility of DO-SRS microscopy in studying the development and metabolic homeostasis of the lipid-rich tissues and to gain new insights into mammalian lipogenesis.

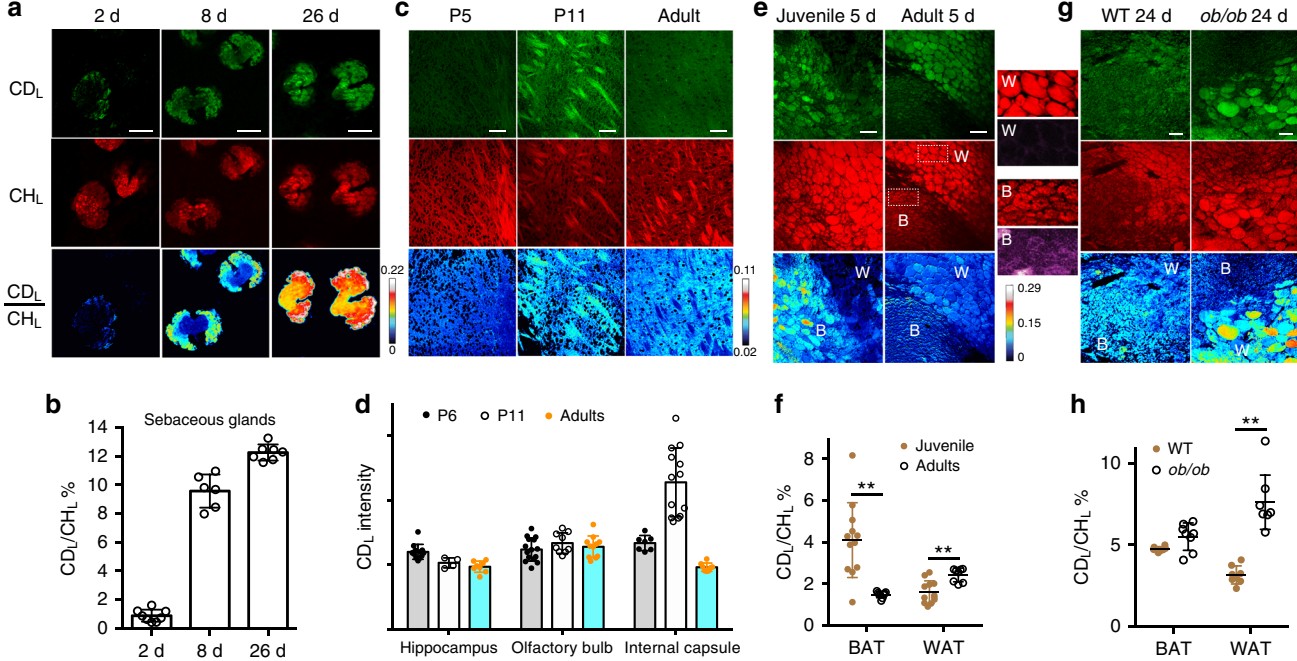

**Fig. 5** DO-SRS microscopy visualizes de novo lipogenesis in mice in vivo. **a, b** Ear skins were harvested from adult mice drinking 25% $D_2O$ for 2, 8, or 26 days, and the sebaceous glands were imaged from the $CD_L$ and $CH_L$ channels, from which signals were color-coded in green and red, respectively. The ratios of $CD_L$ mean intensity to $CH_L$ mean intensity across an entire gland unit were quantified. Mean ± s.d. is plotted; N = 8, 6, and 7 for 2-, 8-, and 26-day treatment, respectively. **c** Internal capsule of the mouse brain from P5 (5 days postnatal) and P11 pups and adults were sectioned and imaged. Pups were fed on milk produced by mother mice drinking 25% $D_2O$ for 6 days before imaging, and adults drank 25% $D_2O$ for 9 days before imaging. **d** Quantification of myelination activities in different brain regions. $CD_L$ signal was quantified as the mean intensity (mean ± s.d.) of the entire image; N is between 8 and 14. **e, f** White and brown adipose tissues from juvenile (P25) and adult (3-month old) mice drinking 25% $D_2O$ for 5 days were imaged. Difference between WAT ('W') and BAT ('B') is shown, as an example, by the enlarged regions (dashed square) of the adult tissues; fluorescence signal excited at 488 nm and collected at 525 nm is shown in purple. **g, h** Adipose tissues from wild-type and *ob/ob* adult mice that drank 25% $D_2O$ for 24 days. For quantification in **f**, **h**, mean ± s.d. is shown. Double asterisks indicate $p < 0.01$ in an unpaired *t* test. Three mice were used for each condition, and three or four fields were imaged for each tissue. Scale bar = 20 μm for the entire figure. The color scale bars represent the scale for $CD_L/CH_L$ ratio

First, we analyzed the spatiotemporal dynamics of lipogenic activities during holocrine secretion. Previous studies imaged the total lipids in sebaceous glands using label-free coherent anti-Stokes Raman scattering microscopy and found that sebocytes near the duct had the highest abundance of lipids[36]. But whether those cells also have the highest lipogenic activity is unclear. By imaging sebaceous glands collected under the ear skin of 3-month-old mice that drank 25% $D_2O$ for 2, 8, or 26 days, respectively, we found that active lipid synthesis occurred mostly in the peripheral sebocytes at day 2, although $CH_L$ images indicated the presence of a large amount of lipids throughout the entire gland (Fig. 5a). The $CD_L$ signal increased and expanded into the center of the gland later at days 8 and 26, as sebocytes migrated toward the duct and accumulated more lipids (Fig. 5a, b). These results suggest that the outermost early sebocytes, instead of the inner mature sebocytes, had the highest lipogenic activity. At day 26, $CD_L/CH_L$ ratio reached ~0.2, which is close to the maximal D:H ratio (0.18–0.21) body water from 15% to 17% $D_2O$ enrichment; thus all the observed lipids were newly synthesized and a likely complete lipid turnover had occurred.

Second, we visualized the myelination dynamics of axon bundles in the developing brain. Previous studies indicated that myelinogenesis occurs predominantly postnatally in mammals[37], but it is rather difficult to obtain the precise timing of myelination in a specific region of the brain with traditional methods. By feeding pups for 6 days (P0–P5 or P6–P11) with milk from mother that drank 25% $D_2O$, we observed strong bundle-like $CD_L$ signal labeling the myelinating thalamocortical fibers in the internal capsule in the second postnatal week (imaged at P11) but not in the first week (imaged at P5) and not in adults (Fig. 5c). Interestingly, previous studies observed organized growth of thalamocortical projection during the first postnatal week[38], and our data suggest that myelination of those fibers occurred shortly after the axonal development within a stringent time window. In contrast, other axonal fiber-rich structures like hippocampus and olfactory bulbs, which mostly developed embryonically, did not show significant change in myelination activities during the same period of time (Fig. 5c, d).

Third, we observed different metabolic dynamics of brown adipose tissue (BAT) and white adipose tissue (WAT) in situ during development and disease. BAT and WAT were identified based on (1) that adipocytes in WAT contain one single, large fat droplet, and adipocytes in BAT contain many small lipid droplets; and (2) that BAT has stronger autofluorescence than WAT due to high levels of cellular NADH and flavin. We found that BAT has higher lipogenic rate ($CD_L/CH_L$ ratio) in juvenile (P25) mice than in adult mice (Fig. 5e, f), consistent with the important thermogenic function of BAT in young animals[39]. WAT serves as energy storage; and age-related increase in the percentage of body fat is often attributed to decrease in resting metabolic rate[40]. However, we found that WAT in adult mice synthesizes and accumulates more fat than WAT in juveniles within the same $D_2O$ probing period, suggesting that increased lipogenesis is also responsible for fat accumulation in adults. Genetically obese (leptin-deficient *ob/ob*) mice also showed much higher lipogenic activities in WAT than did the wild-type mice, indicating that obesity induces ectopic lipogenesis, in addition to fat accumulation (Fig. 5g, h). Hence, DO-SRS microscopy can serve as a

general phenotyping tool for fat metabolism and energy home-ostasis in both normal development and disease.

We also found that $D_2O$ administration via drinking water labeled newly synthesized lipids much more efficiently than the injection of D-FAs. Intravenous injection of 5.76 µmol D-PA into mice over the course of 3 days did not produce significant signal in any tissue. On the other hand, administration of high concentration of D-PA (via intraperitoneal bolus injection of 96 µmol D-PA) led to the accumulation of $CD_L$ signal in the ER membrane of pancreatic cells (Supplementary Fig. 4B), suggesting that, unlike $D_2O$ probing, the D-FA treatment may affect native metabolism in animals by promoting the formation of solid-like domains in the ER, similar to our previous observation in culture cells[30].

**DO-SRS microscopy enables in vivo tracking of de novo protein synthesis**. Protein synthesis activity is another major component of metabolic dynamics. To visualize newly synthesized proteins in mice in vivo, previous studies administered 2.5 mg ml$^{-1}$ of D-labeled amino acids (D-AAs) via drinking water for 12 days and observed CD signals only in liver and intestine tissues but no other organs[41]. This tissue bias may be explained by the unequal uptake of orally administered D-AAs among different organs, heterogeneity of AA pools, and the poor labeling efficiency of D-AAs[26]. Moreover, the possible varying incompleteness of mixing D-AAs with pre-existing AA pool makes the measurement of protein synthesis rate complicated. In contrast, $D_2O$ would have higher labeling efficiency and consistency because $D_2O$ freely diffuses into all tissues and rapidly labels free AAs through transamination[26], making the $CD_P$ signal generated by $D_2O$ probing more accurately reflect the distribution of newly synthesized proteins than the signal from D-AA labeling.

We note that the D-AA concentration (or density) in the free AA pool inside cells (<10 mM)[42] is below our detection limit (~20 mM for singly deuterated molecules), so $CD_P$ signal only reports D-AAs incorporated into proteins. In addition, Busch et al. found that there was no posttranslational labeling of proteins by $D_2O$-derived deuterium[26], thus $D_2O$ probing only tracks protein synthesis and not posttranslational modification.

To demonstrate that $D_2O$ is an efficient, consistent, and cost-effective tracer of de novo protein synthesis, we performed a direct comparison between 8-day administration of 25% $D_2O$ and 2 mg ml$^{-1}$ of D-AAs both via drinking water. Indeed, D-AA treatment only produced significant $CD_P$ signals in the digestive tract and liver, where $D_2O$ probing resulted in much stronger signal (Fig. 6). In the pancreas, D-AA treatment generated weak and sparse signal for protein synthesis activity, whereas $D_2O$ probing was much more sensitive and revealed broader regions of the tissue that actively synthesized proteins. Moreover, in organs, such as hippocampus, cortex, muscle, and thymus, oral intake of D-AAs hardly produced any $CD_P$ signal due to tissue bias, whereas $D_2O$ probing was not subjected to such bias and generated strong $CD_P$ signal labeling newly synthesized proteins. In terms of the cost, $D_2O$ administration (~\$0.67 per mouse per day at 25%) is 15 times cheaper than D-AA treatment (~\$10 per mouse per day at 2 mg ml$^{-1}$), making $D_2O$ a much more economical tracer for long-term in vivo labeling of slow-turnover proteins.

Although non-drinking administration such as injection of D-AAs directly into the bloodstream via carotid artery could enhance CD signals and allow the labeling of proteins in organs, such as pancreas and brain cortex, we found that the signal is still weaker than that generated by $D_2O$ probing. In general, D-AA injection to bloodstream only labels tissues with strong protein synthesis activity and is less sensitive than $D_2O$ treatment, which

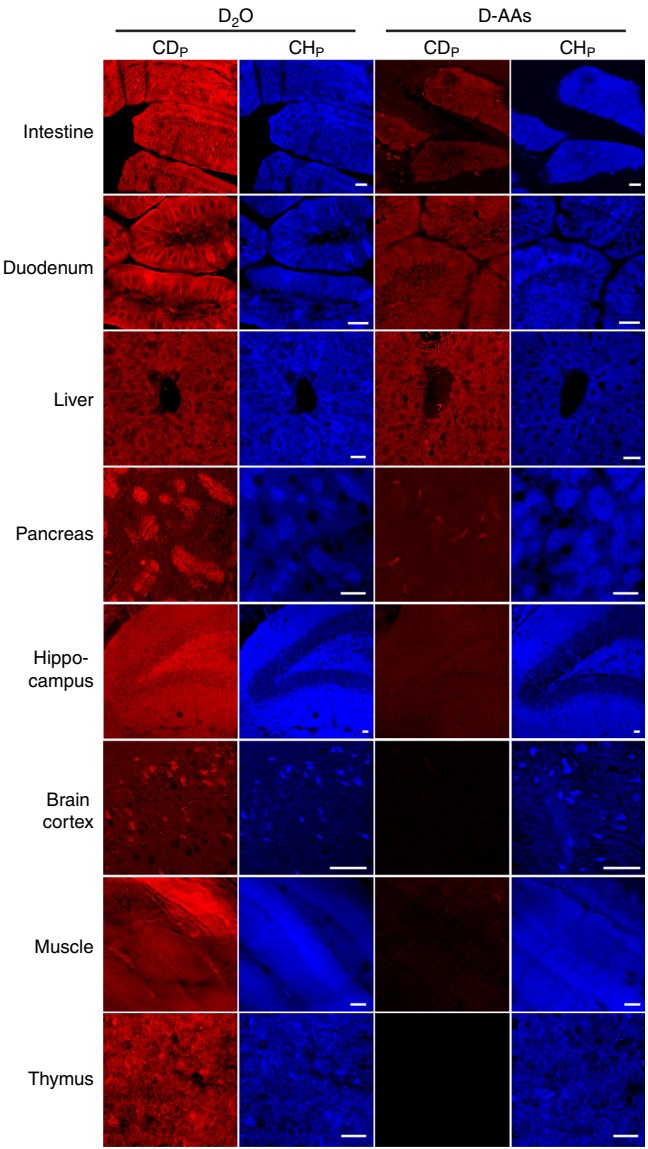

**Fig. 6** DO-SRS microscopy enables tracking of in vivo protein synthesis with high efficiency and without tissue bias. A variety of protein-rich organs were collected from adult mice that were administrated with 25% $D_2O$ or 2 mg ml$^{-1}$ of D-labeled amino acids (D-AAs) in drinking water for 8 days and then imaged for $CD_P$ and $CH_P$ signals. Scale bar = 20 µm

can reveal low levels of de novo protein biosynthesis activities. Furthermore, for organs like hippocampus, injection of D-AAs still could not generate any clear $CD_P$ signal (Supplementary Fig. 4C), suggesting that the D-AA labeling bias against certain tissues is probably inherent to the probe and could not be overcome by increasing D-AA concentration. Compared to the infusion of D-AAs, $D_2O$ administration through drinking water is also much more convenient, enabling large-scale animal experiments.

**Simultaneous visualization of lipid and protein metabolism**. Another major advantage of DO-SRS microscopy, compared to D-FA or D-AA treatment alone, is the ability to simultaneously acquire signals for D-labeled lipids and proteins and then resolve their accurate distribution on the same sample through spectral unmixing, with only one probe ($D_2O$). This unprecedented

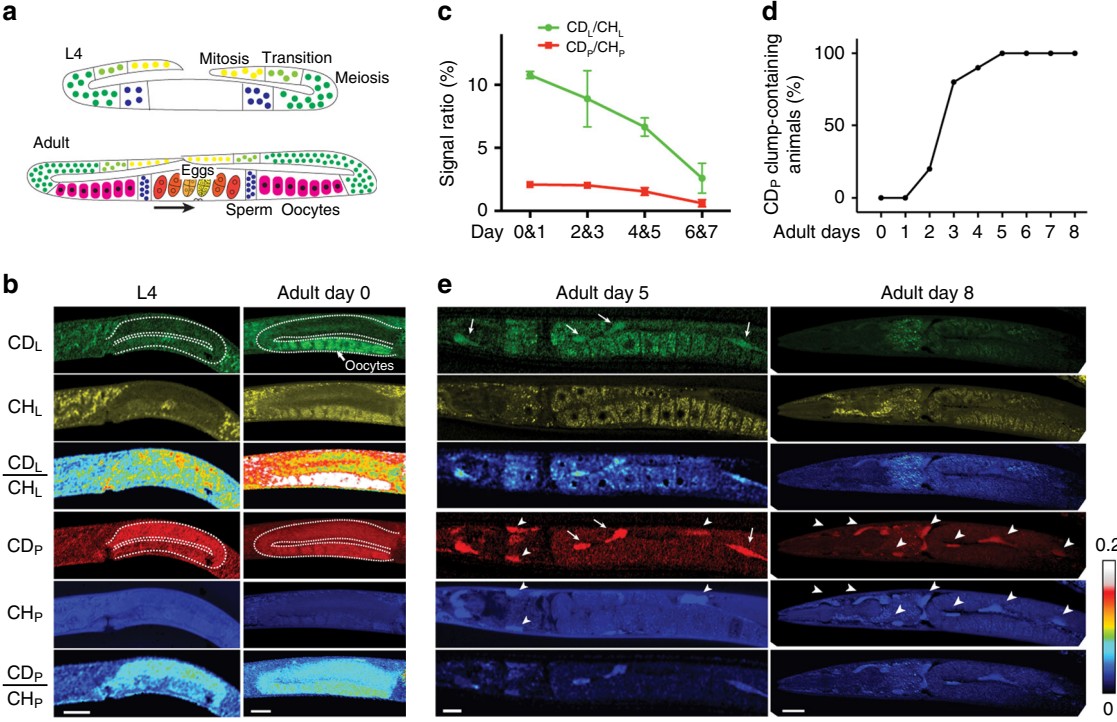

**Fig. 7** DO-SRS microscopy visualizes in vivo protein and lipid metabolism simultaneously in *C. elegans*. **a** Cartoons depict germline development[66] with the following color scheme: yellow mitotic region, light green transition (early prophase of meiosis I), dark green pachytene, dark blue spermatogenesis, and pink oogenesis. In adults, a color gradient from orange to dark yellow indicates the development (arrow) from newly fertilized eggs to 32-cell embryo, which would be expelled via the vulva (triangles). **b** L4 animals and day 0 adults were grown on 20% $D_2O$ plates for 3 h before imaging (dashed outline indicates the gonad). Scale bar = 20 μm. **c** Day 0–7 adults were grown on 20% $D_2O$ plates for 3 h and then imaged. Mean intensity ratios (mean ± s.d.) are shown for animals divided into four age groups. $N = 10$ in each group. **d** Day 0–8 adults were transferred from regular NGM plates to 20% $D_2O$ plates and then imaged in 3 h. The percentages of animals that showed CD signals in clumps in the body cavity and outside of the oocytes are shown; $N = 8$ for each time point. **e** SRS microscopic images of day 5 and day 8 adults after 3 h $D_2O$ probing. Arrows indicate newly formed lipid and protein accumulations only labeled by CD signals, whereas arrowheads indicate pre-existing mass labeled by CH signals. The color scale bars represent the scale for $CD_L/CH_L$ and $CD_P/CH_P$ ratios

capacity allows the study of both lipid and protein metabolism at the same time in an integrated manner.

We first visualized protein and lipid synthesis in *C. elegans* simultaneously during germline development and revealed their different metabolic dynamics. Previous studies suggested that cholesterol, fatty acids, and other nutrients are transported to developing oocytes via yolk particles, but it is unclear when exactly lipid deposition occurs during germ cell development and whether there is any difference between protein and lipid accumulation in oocytes[43]. We found that the mitotic and meiotic germ cells showed very active protein synthesis ($CD_P$) but low level of lipid synthesis ($CD_L$) signals in L4 animals, whereas the post-pachytene, maturing oocytes in day 0 adults accumulated significant amount of newly synthesized lipids in a 3 h period (Fig. 7a, b). Thus our results suggest continuous protein synthesis and accumulation throughout germline development and temporally restricted lipid deposition into late-stage oocytes.

Lipid and protein metabolism also showed different age-related changes in *C. elegans*. In general, overall lipid synthesis rate dropped continuously from adult day 0 to day 7, indicating a decline in lipogenesis during aging. The rate of protein synthesis declined most significantly after adult day 5, showing different dynamics from lipids (Fig. 7c).

The subcellular resolution afforded by DO-SRS microscopy also allowed us to reveal distinct spatial patterns of biosynthesis. Previous studies reported unregulated synthesis and body-wide accumulation of yolk proteins and lipids in the absence of oogenesis in post-reproductive *C. elegans* using either yolk protein::green fluorescent protein (GFP) reporter or electron microscopy (EM)[44,45]. However, fluorescent labeling has high background and cannot differentiate proteins from lipids; EM studies revealed yolk proteins and lipids as different electron-dense materials but are laborious and time-consuming. Importantly, both methods provide only a static view of yolk production, and it is not clear whether the pattern of yolk synthesis changes as the adults age.

Using DO-SRS microscopy, we found that from adult day 3 the majority of worms showed significant $CD_L$ and $CD_P$ signals as clumps in the intestine and throughout the body cavity after a 3 h $D_2O$ probing (Fig. 7d, e). $CD_P$ signal appeared to be stronger than $CD_L$ signal in older adults (e.g., day 8) but not in younger adults (e.g., day 5), indicating more persistent yolk protein synthesis compared to lipogenesis. Our images not only provided a direct visualization of yolk lipid and protein production but also surprisingly revealed some spatial restriction for the previously considered unregulated biosynthesis of yolk materials. For example, we found that $CD_P$ signals appeared in both pre-existing mass and newly formed clumps in younger adults (e.g., day 5), suggesting that newly synthesized proteins both accumulated into existing aggregates and formed new aggregates. However, $CD_P$ signals in older adults (e.g., day 8) only emerged in pre-existing mass, indicating that newly made proteins were only deposited into pre-formed aggregates in aged animals. Thus our observation revealed an aging-dependent aggregation pattern for newly synthesized yolk proteins. DO-SRS microscopy may serve as a useful tool to study protein aggregation in situ.

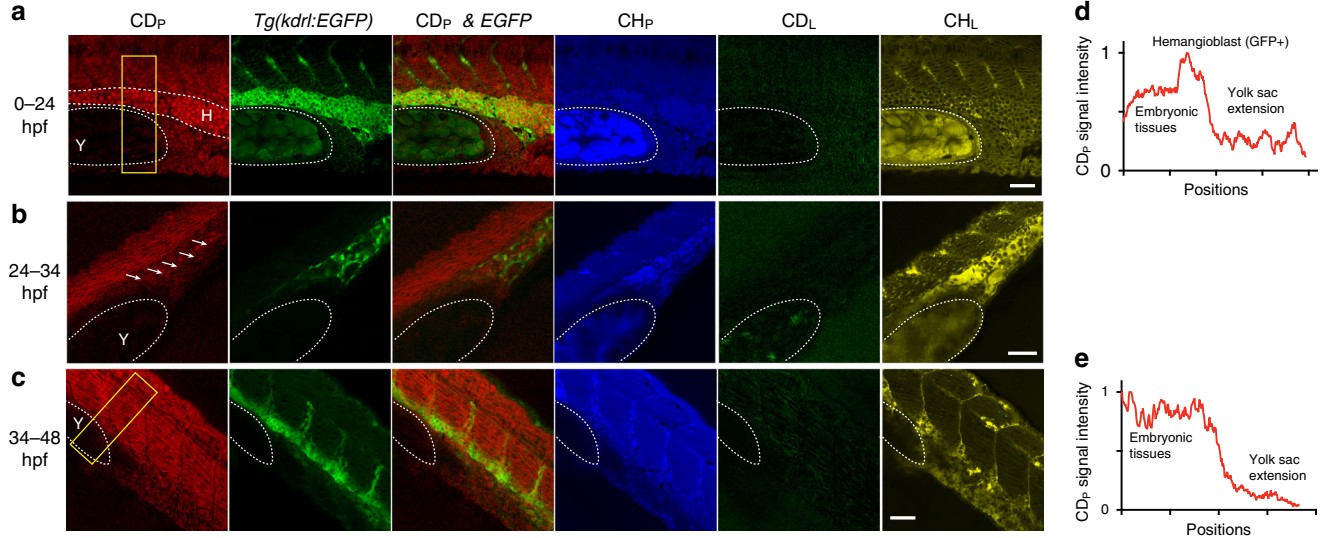

**Fig. 8** DO-SRS microscopy, in combination with fluorescent labeling, tracks lineage-specific metabolism during zebrafish embryogenesis. **a–c** SRS microscopic signal and the colocalization with fluorescence from *Tg(kdrl::EGFP)* reporter in zebrafish embryos that were incubated in egg solution containing 20% $D_2O$ from 0 to 24 hpf, from 24 to 34 hpf, or from 34 to 48 hpf. Dashed curves outline the yolk sac extension (labeled as Y) and the GFP-positive hemangioblast (labeled as H) that has strong $CD_P$ signal. Arrows point to the GFP-positive cells that are likely differentiating angioblasts in **b**. **d**, **e** Intensity profiles that quantify the $CD_P$ signal within the yellow rectangle for 0–24 and 34–48 hpf probing, respectively. *x* axis shows the position along the length of the box from top to bottom, and *y* axis shows the average intensity across the width of the box. Scale bar = 20 μm

Applying this method to zebrafish, we also identified difference in protein and lipid metabolism during embryonic development. By incubating the embryos in egg water containing 20% $D_2O$ for 24 h, we observed significant $CD_P$ signal and little $CD_L$ signal in all embryonically derived tissues, which suggests that the rate of protein synthesis, in general, is much higher than the rate of lipogenesis during embryonic cell division. The zebrafish yolk sac, however, showed very weak signal in both $CD_P$ and $CD_L$ channels despite significant $CH_L$ and $CH_P$ signals from maternally deposited lipids and proteins (Fig. 8). Since the yolk and blastoderm had similar water permeability[46], our result indicated very little zygotic biosynthesis that was independent of maternal contribution in the yolk. Overall, our observations relied on the in situ separation of D-labeled proteins and lipids, which cannot be easily achieved by previous methods.

We also used DO-SRS microscopy in conjunction with fluorescent microscopy to track the metabolic activity of specific cell lineages during development. Probing the embryos from zygote to 24 h post-fertilization (hpf), we found that a group of hemangioblast cells labeled by *Tg(kdrl:EGFP)*[47] had the strongest $CD_P$ signal (Fig. 8a). Those cells originate from lateral plate mesoderm and give rise to endothelial and hematopoietic lineages[48]. Their strong $CD_P$ signal indicates active protein biosynthesis, possibly due to fast proliferation, during the 0–24 hpf period. $D_2O$ probing of later embryonic stages, 24–34 and 34–48 hpf, found that the enhanced GFP (EGFP)-positive cells, which labeled mostly differentiated endothelial cells then[49], no longer showed stronger $CD_P$ signal than the surrounding tissues (Fig. 8b–e). Thus, by co-labeling with lineage-specific fluorescent marker, we can track time-dependent metabolic activities of particular cell types during lineage progression.

**DO-SRS microscopy visualizes tumor boundary and metabolic heterogeneity**. An immediate biomedical application of DO-SRS microscopy is to visualize tumor boundaries and intratumoral metabolic heterogeneity. Although label-free SRS microscopy identified the boundary between glioblastoma and normal brain

tissues[50], it relied on the protein/lipid compositional difference between the tumor and normal tissues—brain areas had higher myelin-derived lipid concentration than the tumor, which may not apply to other types of tumor. In contrast, DO-SRS microscopy can reveal tumor boundaries by tumor's inherently higher metabolic activities than the surrounding normal tissues. For example, giving glioblastoma-bearing mice 25% $D_2O$ for 15 days, we observed both stronger $CD_L$ and $CD_P$ signals in the tumor tissue compared to the nearby brain tissues, even though the brain region had high total lipids in the $CH_L$ channel (Fig. 9a). Unlike the brain tumors, in the subcutaneous xenograft of colon tumor, the tumor and the surrounding skin tissue had similar composition of proteins and lipids and, hence, were indistinguishable by label-free SRS microscopy from their $CH_L$ and $CH_P$ signal. However, through $D_2O$ probing, the tumor showed higher level of lipogenesis than the skin and became readily identifiable in the $CD_L$ channel (Fig. 9b). Thus this example showcases that DO-SRS microscopy could be a more general and applicable method of detecting tumor boundaries than the label-free SRS microscopy.

Intratumoral metabolic heterogeneity is considered a driver of tumor aggressiveness and has been under intensive study due to its fundamental importance as well as prognostic significance[51]. In the colon tumor xenograft, cancer cells recruited stromal cells from the nearby normal tissues and developed into solid tumor. A 3-day $D_2O$ probing revealed that human tumor cells had stronger biogenesis for both proteins and lipids than the recruited mouse stroma, showing metabolic heterogeneity inside the solid tumor (Fig. 9c). Interestingly, the difference between tumor and stroma became less pronounced after 15-day $D_2O$ intake, suggesting that the stromal cells also had significant albeit slower metabolic activity presumably to support tumor growth (Fig. 9d). Therefore, DO-SRS microscopy can visualize the heterogeneity of metabolism inside solid tumors with cellular resolution.

## Discussion

In summary, we developed and demonstrated DO-SRS microscopy as a nondestructive, noninvasive, and background-free

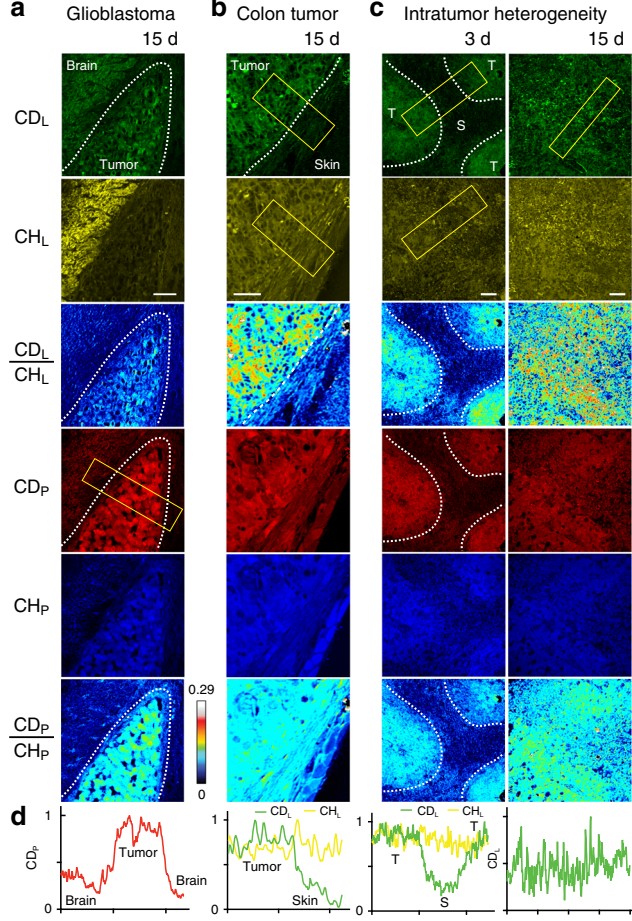

**Fig. 9** DO-SRS microscopy identifies tumor boundaries and metabolic heterogeneity. **a** Intracranial xenograft glioblastoma in mouse brain was excised, sectioned, and imaged after the tumor-bearing mice drank 25% $D_2O$ for 15 days. Color scheme is similar to that in Fig. 2. Dashed curves highlight the tumor–brain boundary visualized by $CD_P$ signal. Intensity profile quantifies the $CD_P$ signal within the yellow rectangle with $x$ axis showing the position along the length of the box and $y$ axis showing the average intensity across the width of the box. **b** Methods similar to **a** were used to visualize the tumor–skin boundary of subcutaneously xenografted colon tumor from the $CD_L$ channel. **c** The interior of the colon tumor xenografts was imaged after the tumor-bearing mice drank 25% $D_2O$ for 3 or 15 days. Dashed lines indicate the boundary between the tumor cells (labeled as T) and the recruited stromal cells (labeled as S), which were identified by the $CD_L$ signals at 3 days. Tumor and stromal cells were also identified by their different morphologies. Scale bar = 20 μm. **d** Intensity profile quantifies the $CD_P$ or $CD_L$ signal within the yellow rectangular with $x$ axis showing the position along the length of the box and $y$ axis showing the average intensity across the width of the box

imaging method that can be used to visualize metabolic dynamics of proteins, lipids, and DNA in a variety of model organisms. DO-SRS microscopy allows for in situ visualization of de novo lipogenesis and protein synthesis in animals at an unprecedentedly low cost and without tissue bias, representing important technical advance. In particular, the ability to simultaneously image newly synthesized lipids and proteins allowed us to gain new insights into the metabolic basis of several biological processes. We also showcased the tremendous potential of using the unmixed $CD_L$ and $CD_P$ signals to identify tumor boundaries and to detect intratumoral heterogeneity.

In this study, we demonstrated that $D_2O$ is a better probe than deuterium-labeled carbon substrate in monitoring and imaging metabolic activities, because $D_2O$ does not perturb native metabolism, can freely diffuse into cells, and is a non-carbon tracer that can probe de novo biosynthesis. For visualizing lipogenic activities, we showed that $D_2O$ is better than D-FAs and also expect $D_2O$ to be better than previously used D-glucose[52], which could potentially create hyperglycemia when used at high concentration and may not label newly synthesized lipids as efficiently as $D_2O$ treatment.

DO-SRS microscopy can be used to address a variety of biological questions. For example, when applied to heterogeneous tumor tissues, DO-SRS microscopy can help identify cancer stem cells that have particular patterns of metabolic activities, such as high lipogenic activities. Label-free SRS microscopy was already used to find that ovarian cancer stem cells had significantly increased levels of unsaturated lipids than non-stem cancer cells[53], and DO-SRS microscopy might be better suited for those studies because of higher sensitivity and the ability to track metabolic dynamics. Another application is to combine DO-SRS microscopy with fluorescent labeling to monitor the metabolic activity of specific cells and lineages in situ. One particularly interesting problem is the metabolic cooperation between glial cells and neurons[54]. In addition to the synthesis of lipids and proteins, DO-SRS microscopy can also be used to monitor protein turnover, lipid consumption, and macromolecule degradation.

Although DO-SRS microscopy has relatively lower molecular specificity and sensitivity compared to mass spectrometric imaging (MSI) methods[55–57], it can serve as an important complementary approach to MSI methods because of several advantages. DO-SRS microscopy provides straightforward and quantitative interpretation of total metabolic activities in 3D living tissues, whereas all MSI techniques are essentially destructive surface analysis, involve more complicated computation than SRS microscopic imaging, and may not capture all D-labeled molecules equally due to certain bias toward easily ionized and desorbed analytes.

A major challenge for in vivo metabolic imaging is the accessibility of tissues deep inside the body. Using devices similar to the coherent Raman scattering endoscopes[58], we envision that DO-SRS microscopy could be applied to visualize metabolic patterns of internal organs and to study tumor metabolism through optical biopsy. The recent development of high-speed, volumetric stimulated Raman projection microscopy and tomography[59] also offers promise in deep-tissue, large-volume, in vivo imaging (e.g., imaging cortical metabolism). Importantly, the sensitivity of our method is high enough to operate in the range of low $D_2O$ enrichment that is safe for humans. Given that SRS microscopic imaging has been demonstrated in humans before[60], we expect DO-SRS microscopy to have clinical application in tracking metabolic activities in humans.

## Methods

**Spontaneous Raman spectroscopy**. Spontaneous Raman scattering spectra are acquired on an upright confocal Raman microspectrometer (Xplora, Horiba Jobin Yvon) with 532 nm diode laser source and 1800 lines per millimeter grating at room temperature. The excitation power is ~40 mW after passing through a 50× air objective (MPlan N, 0.75 N.A., Olympus), and 40 s acquisition time accumulated by 22 was used to collect Raman spectra of all samples at a single point under identical conditions. For cultured cells, the Raman background of water and cover glass is removed from all cell spectra by subtracting the signal at empty space from the signals collected from cells.

**Stimulated Raman scattering microscopy**. We used an inverted laser-scanning microscope (FV1200 MPE, Olympus) optimized for near-infrared (near-IR) throughput and a 25× water objective (XLPlan N, 1.05 N.A., MP, Olympus) with high near-IR transmission for SRS microscopic imaging. A picoEMERALD system

(Applied Physics & Electronics) supplied synchronized pulse pump beam (with tunable 720–990 nm wavelength, 5–6 ps pulse width, and 80-MHz repetition rate) and Stokes (with fixed wavelength at 1064 nm, 6 ps pulse width, and 80 MHz repetition rate). Stokes was modulated at 8 MHz by an electronic optic modulator. Transmission of the forward-going pump and Stokes beams after passing through the samples was collected by a high N.A. oil condenser (N.A. = 1.4). A high O.D. bandpass filter (890/220, Chroma) was used to block the Stokes beam completely and to transmit the pump beam only onto a large area Si photodiode for the detection of the stimulated Raman loss signal. The output current from the photodiode was terminated, filtered, and demodulated by a lock-in amplifier at 8 MHz to ensure shot-noise-limited detection sensitivity. The demodulated signal was fed into analog channel of the FV1200 software FluoView 4.1a (Olympus) to form image during laser scanning.

**Image acquisition parameters**. To acquire CD signals, we set the OPO laser power for the pump beam at 100 mW and the IR laser power for the Stokes beam at 150 mW for all experiments except the *C. elegans* experiments in supplemental Fig. 3A–C, for which we set the OPO laser at 50 mW and the IR laser at 100 mW. To acquire CH signals, we set OPO laser at 100 mW and IR laser at 150 mW except for all *C. elegans* experiments, for which we set the OPO laser at 50 mW and IR laser at 100 mW. The bias voltage of the photodiode detecting the SRS microscopic signal were kept the same as 64 volts for all experiments.

**Cell culture and imaging**. HeLa (ATCC CCL-2), COS-7 (ATCC CRL-1651), and U-87 MG (ATCC HTB-14) cells were obtained from ATCC (Manassas, VA) and cultured in Dulbecco's modified Eagle's medium (DMEM, Thermo Fisher) supplemented with 10% fetal bovine serum (Thermo Fisher) in a humidified 5% $CO_2$ atmosphere at 37 °C. LS174T (ATCC CL-188) was also obtained from ATCC and cultured in Eagle's Minimum Essential Medium (from ATCC). We did not independently authenticate those cell lines and did not test for mycoplasma contamination. To make cell culture medium containing $D_2O$, we used a mixture of $D_2O$ (Sigma-Aldrich, Cat. 151882) and distilled $H_2O$ to dissolve DMEM powder (Sigma) and then sterilized the medium by filtering.

For live-cell imaging, HeLa and COS-7 cells were seeded onto a glass-bottom dish (MatTek) and grown for 6–8 h in regular DMEM before the medium was changed to DMEM containing 70% $D_2O$. Cells were then grown for different periods of time before being placed onto the stage for SRS microscopic imaging. When treating cells with D-labeled fatty acids, they were first reacted with sodium hydroxide and then coupled to BSA in about 2:1 molar ratio to make 2 mM stock solution. Those stock solutions were then added to the medium to reach the final working concentration (10 μM).

To determine cellular toxicity, HeLa cells and COS-7 cells were first seeded in 96-well plates at 500 cm$^{-2}$ and grown overnight; the culture was then changed to DMEM containing different concentration (0–100%) of $D_2O$ for 48 h. The viability of cell was accessed using the CellTiter-Glo® Luminescent Cell Viability Assay (Promega) according to the manufacturer's protocol. We used cells cultured in pure DMEM medium as control and wells containing only medium without cells for background luminescence. Viability was calculated using the background-corrected absorbance as follows: Viability (%) = Absorbance of experiment well/Absorbance of control well × 100%. Three replicates were performed.

**Treatment of inhibitors**. For inhibitor treatments, cells were first grown in glass bottom dishes with DMEM made of $H_2O$ overnight, and then the medium was changed to 70% $D_2O$ DMEM containing either 10 nM fatty acid synthase inhibitor TVB-3166 (Sigma CAS 1533438-83-3) or 1 μM protein synthesis inhibitor anisomycin (Sigma CAS 22862-76-6). Cells were then cultured for 24 h, imaged by SRS microscopy, and fixed by 4% paraformaldehyde for spontaneous Raman spectroscopy.

Methanol wash was performed on fixed cells in glass-bottom dishes and tissues with 99.9% methanol (Sigma) for 30 min and 24 h, respectively, followed by phosphate-buffered saline (PBS) wash. For proteinase K treatment, tissues were first fixed by paraformaldehyde and then treated with 0.2 mg ml$^{-1}$ recombinant proteinase K (Roche) at 37 °C for 15 min and subsequently washed with PBS. Three replicates were performed for those experiments.

**Biochemical extraction of macromolecules**. To extract lipids, HeLa cells were grown in DMEM with 70% $D_2O$ for 24 h, fixed with 4% paraformaldehyde, washed with PBS in petri dishes, and then harvested into 15-ml falcon tubes. In all, 1.3 ml of chloroform and 2.7 ml of methanol were then added to the cells to extract lipids. After centrifugation at 1377 × g for 5 min, the supernatant was transferred to a clean tube. Subsequently, 1 ml of 50 mM citric acid, 2 ml of water, and 1 ml of chloroform were added to the solution and vigorous shaking was used to mix the content. Liquid phases were then separated by centrifugation at 1377 × g for 10 min and the lower phase containing the lipids were transferred to a clean tube. Caps were kept open to allow the evaporation of solvent and the concentration of lipids.

Proteins and DNA were extracted using Trizol (catalog # 15596026, Life Technologies) and DNAzol (catalog # 10503027, Thermo Fisher Scientific), respectively, according to the manuals provided by the manufacturer.

***C. elegans* experiments**. *C. elegans* wild-type (N2) and mutant strains were maintained at 20 °C using nematode growth media (NGM) and *Escherichia coli* strain OP50 as a food source[61]. Different concentrations of $D_2O$ were added to replace $H_2O$ in NGM. *E. coli* OP50 were seeded on $D_2O$-containing NGM plates and grown for 24 h; we then placed worms at different ages onto those plates for various amount of time. To determine the toxicity of $D_2O$, we prepared eggs using hypochlorite, which made the eggshell porous and permeable to $D_2O$, placed them onto plates containing different concentration of $D_2O$, and counted the number of eggs that could not hatch into normal moving larva. We also placed fourth-stage larva onto those plates and counted how many animals became sterile and determined the brood size by counting the total number of hatch larvae from all the eggs laid by one animal. Three replicates were performed.

To prevent deuterium labeling of the bacteria and to feed worms with non-labeled bacteria on $D_2O$ plates, we killed the OP50 bacteria (grown in $H_2O$) immediately after placing them onto $D_2O$ plates with ultraviolet light. Worms grown on dead non-labeled bacteria developed normally and had similar levels of $CH_2$ and $CH_3$ signal, compared to the controls fed on bacteria that grew on $D_2O$ plates for at least 24 h.

To supplement deuterated fatty acids to worms, OP50 bacterial culture was mixed well with 4 mM D31-palmitic acid and then seeded onto NGM plates that contained 100% $H_2O$. As controls, the same OP50 bacteria culture was seeded onto NGM plates that contained 20% $D_2O$. Twenty four hours later, eggs were placed onto those plates and grew until L4 stage before being imaged.

Regular SRS microscopic imaging was done after fixation with 4% formaldehyde for 30 min and washing with PBS. Live imaging was done by directly transferring animals from $D_2O$ plates onto an 4% agarose pad and mount them in M9 solution with 0.1 μm polystyrene beads (Polysciences, Inc.) to reduce mobility. For most *C. elegans* experiments, at least 8 animals were imaged for each condition, and Student's *t* test was used to identify any significant difference between the two treatment groups. No randomization was applied, and the authors were not blinded to the experimental groups.

**Metabolic activity tracking in zebrafish embryos**. Zebrafish embryos carrying the fluorescent reporters *Tg(kdrl:EGFP)*[47] were incubated in egg water[62] supplemented with 20% $D_2O$ for 24 or 48 h from 1-cell stage at 28.5 °C. For $D_2O$ probing from 24 to 36 hpf (hours post fertilization) and from 34 to 48 hpf, embryos were transferred from 100% $H_2O$ environment to egg water containing 20% $D_2O$. After the incubation, embryos were manually dechorionated and fixed in 4% paraformaldehyde overnight at 4 °C. Following fixation, the embryos were transferred to PBS prior to imaging. At least four embryos were imaged for each condition. The zebrafish work was approved by the IACUC (Institutional Animal Care and Use Committee at Columbia University) under protocol AC-AAAJ7554 for ethical regulation.

**$D_2O$ probing in mice**. Wild-type 3–4-month-old adult C57BL/6J and *ob/ob* (B6. Cg-Lep$^{ob}$/J) mice were both obtained from the Jackson Laboratory and were maintained and bred at Columbia University animal facility. For the ex vivo mice tissue experiments, mice that drank 25% $D_2O$ for certain days were anesthetized with isoflurane and sacrificed. Various organs and tissues were harvested, fixed with 4% formaldehyde overnight, and then cut into slices of 120 μm thickness using a vibrating blade microtome (Vibratome, Leica) for SRS microscopic imaging. For the in vivo SRS microscopic imaging of the ear skin, mice were kept anesthetized with isoflurane while one ear was gently sandwiched between a cover slip and a glass slide, which were then placed onto the imaging stage with a heating pad that kept the body warm during the imaging session. For all mice experiments, three mice were used for each condition, and at least three different fields of each tissue were imaged. No randomization was applied, and the authors were not blinded to the experimental groups.

To establish glioblastoma xenograft, we performed intracranial implantation of U-87 MG human glioma cells in nude mice (J:NU). Briefly, the mouse was anesthetized and positioned in stereotaxic instrument (David Kopf Instruments), and then a small section (2 mm in diameter) of the skull was ground with a dental drill until it became soft and translucent. Subsequently, we injected 1.5 × 10$^5$ U87-MG tumor cells (in 3 μl) into the frontal region of the cerebral cortex over the course of 5 min using a 1.5 mm glass capillary. After the implantation, mouse head skin was then closed with SILK sutures (Harvard Apparatus). Two weeks later, mice bearing glioblastoma started and were kept on drinking 25% $D_2O$ for 15 days before being sacrificed for tumor imaging. For the colon tumor xenograft, we injected subcutaneously 1 × 10$^7$ human colorectal LS174T cells into the lower flank of nude mouse. Ten days later, tumor-bearing mice were treated with 25% $D_2O$ as drinking water for 3 or 15 days before being sacrificed for tumor SRS microscopic imaging. The experimental protocols for the mouse studies all complied with ethical regulations and were approved by IACUC under the protocol AC-AAAQ0496.

**Feeding and injection of D-AAs and D-FAs**. To feed mice with D-AAs, D-AA mixture (Catalog # DLM-6819-1, Cambridge Isotope Laboratory) was dissolved in drinking water at 2 mg ml$^{-1}$, and adult mice drank approximately 4 ml of the water every day for 8 days before their organs were harvested for SRS microscopic

imaging. To inject D-AAs into the bloodstream via carotid artery, mice were anesthetized and kept warm on a heating pad. Neck skin of the mouse was cleaned with 2% chlorhexidine solution followed by 70% isopropyl alcohol and the planned incision area was infiltrated subcutaneously with a 1:10 dilution of 50/50 lidocaine (1%). An ~2 cm incision was made along the midline region of the throat to expose the common carotid artery, which was then tied with suture thread (Prolene 86979 Ethicon) on the side closer to heart to stop the blood flow. The artery was then cut with a precision stainless micro-scissor (63041-984, VWR), and a polyurethane based Micro-Renathane catheter tube (MRE033, Braintree Scientific, INC) was carefully cannulated into the opening of carotid artery[63]. A syringe filled with 80 mg ml$^{-1}$ D-AAs dissolved in mammalian Ringer solution was then connected with the cannulated tubing, and liquid flow was controlled by a syringe pump (AL-1000, WPI) to inject the solution at 0.01 ml min$^{-1}$ perfusion rate for half an hour each session for 2.5 days with a time interval of 2 h.

To inject D-PAs intravenously, mice were anesthetized by isoflurane, and one of the carotid arteries was cannulated for the implantation of a catheter. A syringe filled with 4 mM D-PA dissolved in mammalian Ringer's solution was then connected with the implanted catheter, and D-PA solution was perfused at 0.001 ml min$^{-1}$ using a syringe pump (AL-1000, WPI). The administration lasted for 8 h every day for 3 consecutive days before the mice were sacrificed and their organs were harvested, fixed, sectioned, and imaged by SRS microscopy. To inject D-PA intraperitoneally, 0.16 ml of 600 mM D-PA emulsion was injected into the lower right side of abdomen according to a previous method[64]. Seventeen hours later, the mice were sacrificed and their tissues were collected and imaged.

**$CD_L/CD_P/CD_{DNA}$ three-component unmixing**. Three-component unmixing of $CH_L/CH_P/CH_{DNA}$ signals were previously reported[24]. In this study, we developed a similar method to unmix CD signals. Briefly, we first acquired SRS microscopic signals from 2135, 2185, and 2210 cm$^{-1}$ bearing the intrinsic features of lipids, proteins, and DNA, respectively, and then used a linear combination of the three signals with coefficients to determine the amount of the three macromolecules. The signals at 2135 cm$^{-1}$ ($I_{2135}$), 2185 cm$^{-1}$ ($I_{2185}$), and 2210 cm$^{-1}$ ($I_{2210}$) are linear combination of lipid, protein, and DNA concentrations ($CD_L$, $CD_P$, and $CD_{DNA}$) with coefficients $a_L$, $a_P$, $a_{DNA}$, $b_L$, $b_P$, $b_{DNA}$, $c_L$, $c_P$, and $c_{DNA}$ as shown in Eq. 1.

$$\begin{pmatrix} I_{2135} \\ I_{2185} \\ I_{2210} \end{pmatrix} = \begin{pmatrix} a_L & a_P & a_{DNA} \\ b_L & b_P & b_{DNA} \\ c_L & c_P & c_{DNA} \end{pmatrix} \begin{pmatrix} CD_L \\ CD_P \\ CD_{DNA} \end{pmatrix} \tag{1}$$

Unmixing coefficients were obtained from the spectra of D-labeled cellular extracts from $D_2O$-treated HeLa cells (Supplementary Fig. 2A). $a_L$, $b_P$, and $c_{DNA}$ were set to 1, and the rest coefficients were scaled to their relative values. Substituting the coefficients with their values, we obtained Eq. 2 as the following.

$$\begin{pmatrix} I_{2135} \\ I_{2185} \\ I_{2210} \end{pmatrix} = \begin{pmatrix} 1 & 0.38 & 0.29 \\ 0.48 & 1 & 1.08 \\ 0.24 & 0.71 & 1 \end{pmatrix} \begin{pmatrix} CD_L \\ CD_P \\ CD_{DNA} \end{pmatrix} \tag{2}$$

From Eq. 2, we then derived $CD_L$, $CD_P$, and $CD_{DNA}$ as the unmixed D-labeled lipid, protein, and DNA signal intensity, respectively, using Eq. 3.

$$\begin{pmatrix} CD_L \\ CD_P \\ CD_{DNA} \end{pmatrix} = \begin{pmatrix} 1.31 & -0.98 & 0.67 \\ -1.24 & 5.21 & -5.27 \\ 0.56 & -3.47 & 4.58 \end{pmatrix} \begin{pmatrix} I_{2135} \\ I_{2185} \\ I_{2210} \end{pmatrix} \tag{3}$$

**$CD_L/CD_P$ unmixing**. Since unmixed $CD_{DNA}$ signal was very weak in non-dividing cells and in most of the tissues we imaged, we developed a simplified unmixing method that focused on separating $CD_L$ and $CD_P$ signals. We simplified the three-component unmixing equation and removed the $CD_{DNA}$ variable to obtain Eqs. 4, 5, from which we derived the unmixed D-labeled lipid and protein signals

using Eqs. 6 and 7.

$$I_{2135} = a_L \cdot CD_L + a_P \cdot CD_P \tag{4}$$

$$I_{2185} = b_L \cdot CD_L + b_P \cdot CD_P \tag{5}$$

$$CD_L = \frac{a_P \cdot I_{2185} - b_P \cdot I_{2135}}{a_P \cdot b_L - a_L \cdot b_P} \tag{6}$$

$$CD_P = \frac{b_L \cdot I_{2135} - a_L \cdot I_{2185}}{a_P \cdot b_L - a_L \cdot b_P} \tag{7}$$

For $CD_L/CD_P$ unmixing, we measured the coefficiencts using in situ CD lipid and protein signals, which are derived from the spontaneous Raman spectra of pure D-labeled protein signal (signals after 24 h methanol wash) and pure D-labeled lipid signal (signals removed by methanol wash—subtracting methanol-resistant signal from total signal). All signals were first normalized to the phenylalanine peak at 1004 cm$^{-1}$, which stayed constant during methanol wash, and then normalized to the peaks of pure protein and lipid signals, respectively (Supplementary Fig. 2). Thus $a_L$ and $b_P$ were set to 1; the relative intensity of protein bleed-through signal in the lipid channel ($a_P$) and the relative intensity of lipid bleed-through signal in the protein channel ($b_L$) were measured on the spectra. For example, in xenograft tumor tissues (Supplementary Fig. 2C), $a_L = 1$, $a_P = 0.40$, $b_L = 0.51$, and $b_P = 1$.

Thus we obtained the unmixing Eqs. 8, 9 for tumor tissues.

$$CD_L = 1.25 \cdot I_{2135} - 0.50 \cdot I_{2185} \tag{8}$$

$$CD_P = 1.25 \cdot I_{2185} - 0.64 \cdot I_{2135} \tag{9}$$

Using the same method, we also obtained pure D-labeled protein and lipid signals from the mouse pancreas and brain tissues and calculated their specific unmixing coefficients. Data from the three types of tissues were combined to generate a set of average coefficients: $a_L = 1$, $a_P = 0.48 \pm 0.17$, $b_L = 0.42 \pm 0.07$, and $b_P = 1$.

Among the multiple tissues we analyzed, the CD lipid spectrum showed a consistent shape, but CD protein had varying levels of bleed-through into $CD_L$ channel across different tissue types (Supplementary Fig. 2C), which possibly resulted from tissue-specific incorporation of deuterium into various NEAAs and/or different NEAA composition of D-labeled proteins. One extreme example is the pyramidal neurons: their cell body showed almost no SRS microscopic signal at 2135 cm$^{-1}$ but strong signal at 2185 cm$^{-1}$, suggesting almost no protein bleed-through into $CD_L$ channel. Thus, when applying the unmixing algorithm to cells and tissues previously uncharacterized, we tested $a_P$ from 0.31 to 0.65 until the nucleus was deprived of $CD_L$ signal after unmixing. The equations used in this study for $CD_L/CD_P$ signal unmixing of different tissues are presented in Table 1.

**$CH_L/CH_P$ unmixing**. To unmix $CH_L$ lipids and $CH_P$ protein signals, we adapted a previously reported spectral linear combination algorithm[22]. Briefly, we acquired the SRS microscopic signals at 2845 and 2940 cm$^{-1}$, which bear the vibrational features of C–H bonds in lipids and proteins, respectively. The amount of lipids and proteins can then be determined by a linear combination of the signals at those two wavenumbers, with coefficients predetermined by pure substances[22]. Equations 10, 11 were used to calculate unmixed $CH_L$ and $CH_P$ signals. $I_{2845}$ and $I_{2940}$ are SRS microscopic signal intensities at 2845 and 2940 cm$^{-1}$, respectively.

$$CH_L = I_{2940} - I_{2845} \tag{10}$$

$$CH_P = 5 \cdot I_{2845} - 0.4 \cdot I_{2940} \tag{11}$$

**Table 1 Equations used for $CD_L/CD_P$ signal unmixing**

| | $CD_L$ unmixing | $CD_P$ unmixing |
|---|---|---|
| Cos-7 and HeLa cells | $CD_L = 1.25 \cdot I_{2135} - 0.39 \cdot I_{2185}$ | $CD_P = 1.25 \cdot I_{2185} - 0.52 \cdot I_{2135}$ |
| Glioblastoma and colon tumor xenograft | $CD_L = 1.25 \cdot I_{2135} - 0.50 \cdot I_{2185}$ | $CD_P = 1.25 \cdot I_{2185} - 0.64 \cdot I_{2135}$ |
| Zebrafish tissues | $CD_L = 1.25 \cdot I_{2135} - 0.50 \cdot I_{2185}$ | $CD_P = 1.25 \cdot I_{2185} - 0.52 \cdot I_{2135}$ |
| C. elegans | $CD_L = 1.25 \cdot I_{2135} - 0.70 \cdot I_{2185}$ | $CD_P = 1.25 \cdot I_{2185} - 0.52 \cdot I_{2135}$ |
| Mouse sebaceous glands and adipose tissues | $CD_L = 1.25 \cdot I_{2135} - 0.39 \cdot I_{2185}$ | $CD_P = 1.25 \cdot I_{2185} - 0.81 \cdot I_{2135}$ |
| Mouse internal capsule and cortex | $CD_L = 1.25 \cdot I_{2135} - 0.39 \cdot I_{2185}$ | $CD_P = 1.25 \cdot I_{2185} - 0.70 \cdot I_{2135}$ |
| Mouse liver, pancreas, and muscle tissues | $CD_L = 1.25 \cdot I_{2135} - 0.35 \cdot I_{2185}$ | $CD_P = 1.25 \cdot I_{2185} - 0.60 \cdot I_{2135}$ |

**Image processing**. We used the Olympus FluoView 4.1a scanning software to acquires images and ImageJ to assign color, calculate ratiometric values, and overlay images. All SRS microscopic image were generated by subtracting non-resonance background from the resonance signal. For the in vivo imaging of sebaceous glands in living mice (Fig. 1d and Supplementary Movies 1–3), the two-photon absorption by blood cells is independent of the resonance state, and the fast dynamics of the signal in the blood flow distinguishes it from the more static SRS microscopic signal.

$CD_L/CH_L$ and $CD_P/CH_P$ ratios were used to show the proportion of newly synthesized lipids and proteins to total macromolecules, as an indication of relative metabolic rate. For example, $CD_L/CH_L$ ratio can be calculated using Eq. 12.

$$\frac{CD_L}{CH_L} = \frac{a*[C-D]}{b*[C-H]} = \frac{a*[\text{new lipid}]*G}{b*([\text{new lipid}]*(1-G)+[\text{old lipid}])} \quad (12)$$

[C – D] and [C – H] are the concentrations of D-labeled and H-labeled lipids, respectively; $a$ and $b$ are converting factors from lipid concentrations to SRS microscopic signals intensity; [old lipid] is the amount of lipids that pre-existed before $D_2O$ probing and remained at the time of imaging (not being degraded); [new lipid] is the amount of newly synthesized lipids; and $G$ is the proportion of D-labeled lipids to the newly synthesized lipids.

If we define [total lipid] as the amount of total lipid after the probing period and at the time of imaging, [total lipid] = [new lipid] + [old lipid], then Eq. 10 can be rewritten as Eq. 13.

$$\frac{CD_L}{CH_L} = \frac{a}{b} \cdot \frac{G}{\frac{[\text{total lipid}]}{[\text{new lipid}]} - G} \quad (13)$$

Since the upper limit of $G$ is ~0.2 ($D_2O$ enrichment) in all our experiment, much smaller than [total lipid]/[new lipid], the $CD_L/CH_L$ can be approximated using Eq. 14.

$$\frac{CD_L}{CH_L} = \frac{a}{b} \cdot G \cdot \frac{[\text{new lipid}]}{[\text{total lipid}]} \quad (14)$$

Thus $CD_L/CH_L$ is linear to the proportion of the amount of newly synthesized lipids to total lipids and therefore a ratiometric measurement of lipid synthesis rate.

**Data availability**. All data generated or analyzed during this study are included in this published article and supplementary information file.

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

## Acknowledgements

We thank Dr. Martin Chalfie for critical comments on the manuscript. C.Z. is supported by NIGMS grants (GM30997 and GM122522) to Martin Chalfie. C.d.S.T and K.T. are supported by NHLBI R01 (HL131438-01A1). W.M. acknowledges support from the NIH Director's New Innovator Award (1DP2EB016573), R01 (EB020892), the US Army Research Office (W911NF-12-1-0594), the Alfred P. Sloan Foundation, and the Camille and Henry Dreyfus Foundation.

## Author contributions

L.S., C.Z., Y.S, Z.C, and W.M. conceived the project. L.S., C.Z., Y.S., E.S.S, L.Z., M.W., C.L., and C.d.S.T. performed experiments. C.Z. and L.S. wrote the original manuscript. C.Z., L.S., Y.S., W.M., Z.C., C.d.S.T., and K.T. edited the manuscript. K.T. and W.M secured funding.

## Additional information

**Competing interests:** The authors declare no competing interests.

