## [Peer Review File · Nature Communications]

Reviewers' comments:

Reviewer #1 (Remarks to the Author):

The manuscript entitled "Optical Imaging of Metabolic Dynamics in Animals" by Shi et al. details a massive study of the incorporation of deuterium from D₂O into various macromolecules, compartments/organelles and organs of living cells, *C. elegans*, and mice. The paper provides a number of significant new observations and results concerning the metabolic uptake and conversion of deuterated water into cells and tissue, which absolutely support the publication of this paper. While the results are significant, the sheer number of data is somewhat overwhelming and I believe that less might have been more in this case. I certainly applaud the authors to their success and the tremendous amount of work and effort that has gone into this work, but there are a few items that I think still require correction before I can fully support the publication of this work:

1) Language: the paper is at times difficult to read/understand because it contains grammatical errors, such as many missing articles, which are somewhat irritating. The issues are too frequent to list them all, so I highly recommend that the authors either ask a native speaking person to read and correct the paper or that they utilize an external proof-reading service.

2) The specific type of incorporation of deuterium into newly synthesized lipids is rather speculative. While it appears obvious that often just a single (sparse) H is being replaced by a D, the specific location of this incorporation/labeling is not truly supported by the results shown in this paper. I fully agree that this is a particularly difficult experiment to perform and it would warrant an entirely separate study by itself. I think that removing this speculative part from figure 1, or clearly highlighting the speculative nature in the figure and figure caption is required.

3) Reproducibility: in order for others to reproduce and check this work, additional information about the specific conditions at which the SRS experiments were conducted, is required, such as: - each figure / subfigure should clearly state the power or pump and Stokes beams at which the data were acquired. Was the bias voltage of the photodiode detecting the SRS signal kept the same for all experiments or was it adjusted? If so, the bias voltage should be listed as well.

4) Figure 1: Part D of this figure supposedly shows blood flow based on two-photon absorption by blood cells. How was this result obtained and how can it be separated from the SRS imaging of the gland itself? With the current figure, the entire figure is shown with one color, implying that the TP absorption by blood is an intrinsic signal that was imaged simultaneously with the SRS signal, so how can you be certain about the true nature of each of the signals stemming from different parts of this image?

5) Figure 2: In the caption the "spectra" are labeled as "spontaneous Raman spectra". It should be noted that only a very small part of an overall Raman spectrum is shown for these components. So, rather than labeling these as "spectra" I would suggest that authors rephrase this as either "high wavenumber spectra" or the "deuterium molecular group related part of the Raman spectrum", or any other, more specific phrase that the authors can come up with.

As a final note: I couldn't find a note confirming IRB board approval of the various animal experiments, which I think is a policy of any Nature journal, so this needs to be added to the paper.

Reviewer #2 (Remarks to the Author):

In the manuscript, Shi et al. introduced a technique combining D₂O probing and stimulated Raman

scattering microscopy (DO-SRS) to image in situ metabolic activities. This work is another new development based on the team's strong track record in developing novel analytical methods with Raman techniques. Authors also provided a wide range of example applications of this technique to demonstrate its capabilities in lipid and protein metabolism studies. In general, this is a well-executed study. The technique developed by authors is novel to my knowledge. Live imaging on transparent animal models with DO-SRS is impressive.

Major concerns:

The authors should include discussions of the limitations of this technique.

Some of the applications provided by authors are done on tissue sections. It's hard to justify the nondestructive and noninvasive advantages claimed by authors on applications on tissue sections. Maybe the authors should be more focused on the studies on live animals.

The authors have demonstrated that DO-SRS can be used in a wide range of organisms or tissues, but it would be more convincing for readers if authors can provide one or two specific biological hypothesis that can be studied using this technique in the Discussion section.

Minor points:

What's the earliest time point that authors can detect the C-D signal? How different of this for CDL and CDP?

Authors made a good point that D-FAs may change the native cellular metabolism for cell culture, but it may not be evident for animal experiments. Maybe the authors should combine the comparisons between D₂O or D-FA probing.

Would D-FAs or D-AAs provide a higher signal at earlier time points and be useful for some studies?

The authors could include discussions of what percentage of the protein synthesis is from de novo synthesized amino acids if available. This would be also useful to justify the comparison between the administration of 25% D₂O and administration of D-AAs via drinking water or injection.

As I understand, MSI methods don't need intensive computation to achieve similar information can be achieved from DO-SRS. The bias ionization of analytes also doesn't affect the measurement of total newly synthesised molecules as shown by CD/CH from DO-SRS.

Reviewer #3 (Remarks to the Author):

In this work, the authors map the de novo synthesis of tissue components after feeding animals (or cells) D₂O. Although the principle of using D₂O for measuring de novo synthesis of lipids, proteins and other components is not new, the implementation here is truly of a different kind compared to previous publications. The scale and amount of detail that the authors present here is impressive. The results are inspiring, and I have no doubt that this work will spark a renewed interest in D₂O-based 'labeling' of tissue components. Because of the scope and beautiful demonstrations provided in this work, I strongly recommend this manuscript for publication.

A couple of suggestions for points of discussion:

1) The authors present impressive results on de novo lipid synthesis. They discuss the advantages of D₂O probing with previous work that used deuterated fatty acids. It is clear that D₂O probing has advantages over the previous method. However, a related method, namely the use of deuterated glucose, is not discussed here. D-glucose gets metabolized through a well documented

cascade of reactions, including the conversion of pyruvate to acetyl-CoA, the main precursor to all fatty acids that are used for neutral lipid synthesis. Previous SRS imaging has shown the feasibility of studying lipogenesis through the use of D-glucose. Unlike D₂O probing, which incorporates C-D bonds at any stage of the cascaded reactions, the use of D-glucose enables a single precursor from which the lipids emanate. The latter aspect may have advantages over D₂O probing. A discussion would be useful.

2) The authors focus on the appearance of the C-D signatures as a marker of de novo synthesis. However, one could also focus on the disappearance of such signatures, as a marker for protein lifetime, chemical turnover rates or lipid consumption rates. A short discussion or perspective on the depletion of C-D signals would be useful.

Reviewer #1 (Remarks to the Author):

The manuscript entitled "Optical Imaging of Metabolic Dynamics in Animals" by Shi et al. details a massive study of the incorporation of deuterium from D₂O into various macromolecules, compartments/organelles and organs of living cells, *C. elegans*, and mice. The paper provides a number of significant new observations and results concerning the metabolic uptake and conversion of deuterated water into cells and tissue, which absolutely support the publication of this paper. While the results are significant, the sheer number of data is somewhat overwhelming and I believe that less might have been more in this case. I certainly applaud the authors to their success and the tremendous amount of work and effort that has gone into this work, but there are a few items that I think still require correction before I can fully support the publication of this work:

Response: We much appreciate the strong support and enthusiasm of Reviewer 1.

1) Language: the paper is at times difficult to read/understand because it contains grammatical errors, such as many missing articles, which are somewhat irritating. The issues are too frequent to list them all, so I highly recommend that the authors either ask a native speaking person to read and correct the paper or that they utilize an external proof-reading service.

Response: We have carefully proofread the manuscript and corrected some errors and mistakes.

2) The specific type of incorporation of deuterium into newly synthesized lipids is rather speculative. While it appears obvious that often just a single (sparse) H is being replaced by a D, the specific location of this incorporation/labeling is not truly supported by the results shown in this paper. I fully agree that this is a particularly difficult experiment to perform and it would warrant an entirely separate study by itself. I think that removing this speculative part from figure 1, or clearly highlighting the speculative nature in the figure and figure caption is required.

Response: We added a sentence to the legend of Figure 1 to inform the readers that this illustration only represents one possible deuterium labeling pattern and the position of D replacement of H would be random under the condition of sparse labeling.

3) Reproducibility: in order for others to reproduce and check this work, additional information about the specific conditions at which the SRS experiments were conducted, is required, such as: - each figure / subfigure should clearly state the power or pump and Stokes beams at which the data were acquired. Was the bias voltage of the photodiode detecting the SRS signal kept the same for all experiments or was it adjusted? If so, the bias voltage should be listed as well.

Response: We added the information about the laser power in the materials and methods section, under a new subsection called "Image acquisition parameters". In general, we used the same laser power (100 mw for pump beam and 150 mw for Stokes beam on sample) for all the experiments except for a few *C. elegans* experiments. The bias voltage of the photodiode was kept the same as 60 volts for all experiments.

4) Figure 1: Part D of this figure supposedly shows blood flow based on two-photon absorption by blood cells. How was this result obtained and how can it be separated from the SRS imaging of the gland itself? With the current figure, the entire figure is shown with one color, implying that the TP absorption by blood is an intrinsic signal that was imaged simultaneously with the SRS signal, so how can you be certain about the true nature of each of the signals stemming from different parts of this image?

Response: SRS signal is generated by subtracting the non-resonance background from the resonance signal, whereas the two-photon absorption by blood cells shows signal independent of the resonance state. To support this point, we added movies with resonance off for the sebaceous glands in living mice as Movie S3, which clearly shows the blood flow but no SRS signal.

With regards to Figure 1D, because of the fast dynamics of the blood flow, two-photon absorption in the blood can still be seen after we subtract resonance off background from resonance on signal, thus showing the same color as the SRS signal. However, the blood signal change from frame to frame, distinguishing it from the more static SRS signal of the gland. Moreover, the shape of the blood vessel and the persistence of the signal in the resonance off image help us separate the two-photon absorption of the blood cells from real SRS signal.

To clarify these technical details, we added more explanation in the Methods about imaging acquisition and processing.

5) Figure 2: In the caption the "spectra" are labeled as "spontaneous Raman spectra". It should be noted that only a very small part of an overall Raman spectrum is shown for these components. So, rather than labeling these as "spectra" I would suggest that authors rephrase this as either "high wavenumber spectra" or the "deuterium molecular group related part of the Raman spectrum", or any other, more specific phrase that the authors can come up with.

Response: We changed “Spontaneous Raman spectra” to “High wavenumber Raman spectrum (2030 to 2330 cm^{-1})” in the legend of Figure 2.

As a final note: I couldn't find a note confirming IRB board approval of the various animal experiments, which I think is a policy of any Nature journal, so this needs to be added to the paper.

Response: Animal experiments in this study were approved by IACUC (the Institutional Animal Care and Use Committee at Columbia University). We have disclosed the IACUC-approved protocol number (AC-AAAJ7554 for the zebrafish work and AC-AAAQ0496 for the mouse studies) in the methods section.

Reviewer #2 (Remarks to the Author):

In the manuscript, Shi et al. introduced a technique combining D₂O probing and stimulated Raman scattering microscopy (DO-SRS) to image *in situ* metabolic activities. This work is another new development based on the team's strong track record in developing novel analytical methods with Raman techniques. Authors also provided a wide range of example applications of this technique to demonstrate its capabilities in lipid and protein metabolism studies. In general, this is a well-executed study. The technique developed by authors is novel to my knowledge. Live imaging on transparent animal models with DO-SRS is impressive.

Response: We much appreciate the strong support and enthusiasm of Reviewer 2.

Major concerns:

The authors should include discussions of the limitations of this technique.

Response: We added more discussion about the limitations of DO-SRS in the discussion section. In our opinion, DO-SRS is a powerful tool to image general metabolic activity *in situ* in living tissues, but it currently has limited specificity, only being able to identify major types of macromolecules. We are in the process of using hyperspectral analysis to extract information about specific types of lipids from the C-D signal. With the improvement in imaging techniques and image processing methods, it may be possible to identify spectral signature associated with specific molecules (e.g. D-labeled saturated fatty acids, D-cholesterol, etc) in the broad C-D spectrum, thus increasing the specificity of DO-SRS.

Some of the applications provided by authors are done on tissue sections. It's hard to justify the nondestructive and noninvasive advantages claimed by authors on applications on tissue sections. Maybe the authors should be more focused on the studies on live animals.

Response: We had to section some mouse tissues to examine their C-D signal, because those internal organs of mice are not accessible by any live imaging methods to our knowledge. Two-photon microscopy and SRS microscopy have an imaging depth of ~ 1mm, and stimulated Raman scattering endoscopes is not available in our lab yet (although a prototype of coherent Raman scattering endoscopes was reported, the technology is not mature enough yet). Thus, SRS imaging of organs like liver and pancreas in living animals is not quite possible.

In order to establish an *in vivo* labeling method, we think it is important to examine the signal for D-labeled macromolecules in many tissue types from multiple organs and thus included the work on tissue sections. Nevertheless, we envision that with the advancement in multi-photon microscopy and Raman endoscopy, it may become possible, in the near future, to image deep inside the body and internal organs in living animals. Our current work on the distribution of C-D signal patterns in various tissues would provide reference for those future efforts.

Moreover, in addition to living imaging of animals, the nondestructive and noninvasive advantages also include the scenarios where the samples after SRS imaging can be later examined by other imaging modalities.

The authors have demonstrated that DO-SRS can be used in a wide range of organisms or tissues, but it would be more convincing for readers if authors can provide one or two specific biological hypothesis that can be studied using this technique in the Discussion section.

Response: We added a few biological questions that can be studied using DO-SRS in the discussion section. For example, applying DO-SRS to tumor tissues, one may discover cancer stem cells that have particular patterns of metabolic activities, such as high lipogenic activities, and then identify molecular markers for those cells by comparing the C-D images with biochemical labeling images. Supporting the feasibility of this proposal, Li *et al* (2017; PMID: 28041894) recently used label-free SRS to find that ovarian cancer stem cells had significantly increased levels of unsaturated lipids than other cells in the tumor. DO-SRS will be better suited to study metabolic heterogeneity than label-free SRS because of higher sensitivity and the ability to track dynamics.

Other application of this technique includes combining it with conventional fluorescent labeling to track the metabolic status of specific cells and lineages during development and studying the metabolic dynamics of tissues in a variety of disease models, such as diabetes.

Minor points:

What's the earliest time point that authors can detect the C-D signal? How different of this for CD_L and CD_P?

Response: In cells treated with medium containing 70% D₂O, the earliest time point we can detect weak C-D signal is around 30 minutes after the change of medium. In animal experiments, the timing of the emergence of C-D signal is dependent on specific organs and the age of the animals. For example, in juvenile mice, C-D signals in sebaceous glands appeared one day after feeding the animals with 25% D₂O drinking water, whereas the signal appeared two days after the treatment in adult mice. Organs like muscles, kidney, and brain require much longer time (~7 days) to accumulate detectable CD signals. In general, we observed the first appearance of CD_L and CD_P signal at roughly the same time, although depending on the tissue type, one of the two signals will increase more dramatically than the other over time.

Authors made a good point that D-FAs may change the native cellular metabolism for cell culture, but it may not be evident for animal experiments. Maybe the authors should combine the comparisons between D₂O or D-FA probing. Would D-FAs or D-AAs provide a higher signal at earlier time points and be useful for some studies?

Response: D-FA is very inefficient in labeling lipids in animal experiments. We intravenously injected 5.76 μmol of D-palmitic acid (D-PA) emulsion into mice over the course of three days and did not observe significant signal in any tissue. Intraperitoneal bolus injection of 96 μmol D-PA led to some CD_L signal in the pancreas 17 hours after the injection, but the signal pattern looks quite different from the pattern generated by D₂O probing. It appears that this short-term D-PA probing induced strong C-D signals in the ER membrane (new supplemental Figure 4B), which was not observed even in animals that were administered with 25% D₂O for 8 days. Moreover, those C-D signals in the ER membrane looked similar to the abnormal solid-like domains in the ER induced by high concentration of PA in cells (Shen *et al*, 2017; PMID: 29196526). Thus, like the observations we made in cultured cells, D-FA treatment may also change native metabolism in animal experiments.

Nevertheless, D-FA may be better suited than D₂O for the study of the metabolism of saturated fatty acids. That will be a specific metabolic process rather than the more general lipid metabolism.

The authors could include discussions of what percentage of the protein synthesis is from de novo synthesized amino acids if available. This would be also useful to justify the comparison between the administration of 25% D₂O and administration of D-AAs via drinking water or injection.

Response: Our results suggests that in the case of D₂O probing, most of the deuterium labeling in newly synthesized proteins occurred at the α carbon position through reversible transamination of free AAs, which suggests that D₂O-derived deuterium can potentially label both essential and nonessential amino acids, and this labeling does not require *de novo* synthesis of amino acids.

How much D-AAs can be incorporated into newly synthesized proteins depends on the amount of free AAs in the tissue (which may depend on the nutritional status of the animal), the level of cellular uptake of D-AAs, and the mixing of D-AAs with the endogenous AA pools. Thus, it is difficult to predict how much D-AAs are needed to generate similar level of protein labeling as D₂O treatment. In practice, it is cost-inhibitive to administer high concentration of D-AAs to mice *via* drinking water for a long period of time. Compared to D-AAs, D₂O is a much more efficient tracer for *in vivo* protein synthesis and does not have tissue bias.

As I understand, MSI methods don't need intensive computation to achieve similar information can be achieved from DO-SRS. The bias ionization of analytes also doesn't affect the measurement of total newly synthesised molecules as shown by CD/CH from DO-SRS.

Response: We have changed those statements in the discussion to more accurately reflect the difference between DO-SRS and MSI. Although MSI method may not require intensive computation, SRS imaging processing is still much easier and more straightforward by comparison. The bias towards easily ionized and desorbed analytes means that signals from D-labeled molecules may not equally be captured by MSI, but SRS does not require ionization and desorption and, hence, can collect all signal from D-labeled macromolecules *in situ*. Nevertheless, DO-SRS may have lower sensitivity and specificity than MSI methods.

Reviewer #3 (Remarks to the Author):

In this work, the authors map the de novo synthesis of tissue components after feeding animals (or cells) D₂O. Although the principle of using D₂O for measuring de novo synthesis of lipids, proteins and other components is not new, the implementation here is truly of a different kind compared to previous publications. The scale and amount of detail that the authors present here is impressive. The results are inspiring, and I have no doubt that this work will spark a renewed interest in D₂O-based 'labeling' of tissue components. Because of the scope and beautiful demonstrations provided in this work, I strongly recommend this manuscript for publication.

Response: We much appreciate the strong support and enthusiasm of Reviewer 3.

A couple of suggestions for points of discussion:

1) The authors present impressive results on de novo lipid synthesis. They discuss the advantages of D₂O probing with previous work that used deuterated fatty acids. It is clear that D₂O probing has advantages over the previous method. However, a related method, namely the use of deuterated glucose, is not discussed here. D-glucose gets metabolized through a well-documented cascade of reactions, including the conversion of pyruvate to acetyl-CoA, the main precursor to all fatty acids that are used for neutral lipid synthesis. Previous SRS imaging has shown the feasibility of studying lipogenesis through the use of D-glucose. Unlike D₂O probing, which incorporates C-D bonds at any stage of the cascaded reactions, the use of D-glucose enables a single precursor from which the lipids emanate. The latter aspect may have advantages over D₂O probing. A discussion would be useful.

Response: We added a new reference to the study by Li and Cheng (2014) who directly visualized *de novo* lipogenesis in cultured cells using D-glucose and SRS. Compared to D₂O, D-glucose is still a nutritional supplement and can potentially perturb the native metabolism of cells and animals. In Li and Cheng's study, 25 mM D7-glucose was added to the medium to generate sufficient signal, whereas the physiological level of glucose in the blood is about 5 mM. The fact that D-glucose could only label fatty acids through the conversion to D-labeled acetyl-CoA suggests potentially low labeling efficiency, thereby requiring a high concentration of D-glucose.

We found that animals fed with 5% D-glucose in drinking water produced CD_L signal that has only ~1/4 the intensity of the signal generated in animals that drank 25% D₂O (unpublished results). Moreover, D-glucose is significantly more expensive than D₂O. Thus, when labeling the total metabolic activities, D₂O probing is much cheaper and more efficient than probing with D-glucose. Nevertheless, one potential advantage of using D-glucose over D₂O is that one could label glucose with deuterium at different carbon positions and use those deuterated isomers to achieve distinct labeling patterns of macromolecules *in vivo*. We plan to report those results in a separate paper elsewhere.

2) The authors focus on the appearance of the C-D signatures as a marker of de novo synthesis. However, one could also focus on the disappearance of such signatures, as a marker for protein lifetime, chemical turnover rates or lipid consumption rates. A short discussion or perspective on the depletion of C-D signals would be useful.

Response: We have actually showed that the disappearance of C-D signal can be used to monitor the turnover of lipids in *C. elegans* through a pulse-chase experiment (Figure 4E; results were described in the third paragraph on Page 9). As the reviewer suggested, we added more discussion about the potential use of DO-SRS to study the degradation and turnover of macromolecules.

REVIEWERS' COMMENTS:

Reviewer #1 (Remarks to the Author):

The authors have fully addressed all of my initial concerns, so I now support the publication of this latest iteration as is.

Reviewer #2 (Remarks to the Author):

I support the publication of this manuscript with a very minor revision.

I very much appreciate that the authors added a new figure from samples treated with D-FAs. However, I don't agree with the comparisons between D2O and D-FAs authors included. I agree that D2O has some advantages over D-FAs, but the reason that the D-FAs changed the native metabolism is likely due to the way they administered the D-FAs (intraperitoneal injection). The authors should change the comparison to specifically about the differences between these two treatments (intraperitoneal injection of D-FAs and drinking D2O), rather than generalise it as differences between the two probes.

Another minor issue: most MSI techniques don't use fixed samples.

I am happy with other response and edits authors made in the manuscript.

REVIEWERS' COMMENTS:

Reviewer #1 (Remarks to the Author):

The authors have fully addressed all of my initial concerns, so I now support the publication of this latest iteration as is.

Response: We appreciate the reviewer's support for this study.

Reviewer #2 (Remarks to the Author):

I support the publication of this manuscript with a very minor revision.

Response: We appreciate the reviewer's support for publication.

I very much appreciate that the authors added a new figure from samples treated with D-FAs. However, I don't agree with the comparisons between D₂O and D-FAs authors included. I agree that D₂O has some advantages over D-FAs, but the reason that the D-FAs changed the native metabolism is likely due to the way they administered the D-FAs (intraperitoneal injection). The authors should change the comparison to specifically about the differences between these two treatments (intraperitoneal injection of D-FAs and drinking D₂O), rather than generalise it as differences between the two probes.

Response: We have removed the statement “Because D₂O does not perturb endogenous metabolism, it is also a better probe than D-FAs for tracking lipogenic activities in animal experiments” in the Results and specified the comparison is made between D₂O administration through drinking water and D-FAs through intraperitoneal injection.

Another minor issue: most MSI techniques don't use fixed samples.

Response: We have removed the phrase “(all MSI techniques) can only be applied to fixed tissues.”

I am happy with other response and edits authors made in the manuscript.